# Insights into tectonic zonation models from the clustering analysis of seismicity in South and South-eastern Spain

David Montiel-López[1], Antonella Peresan[2], Elisa Varini[3], and Sergio Molina[1,4]

[1]Multidisciplinary Institute for Environmental Studies "Ramón Margalef" (IMEM), University of Alicante, Ctra. San Vicente del Raspeig, s/n, 03080 Alicante, Spain
[2]National Institute of Oceanography and Applied Geophysics (OGS), Sgonico, Trieste, Italy
[3]Institute for Applied Mathematics and Information Technologies (IMATI), National Research Council (CNR), Milan, Italy
[4]Department of Applied Physics, University of Alicante, Ctra. San Vicente del Raspeig, s/n, 03080 Alicante, Spain

**Correspondence:** David Montiel-López (david.montlop@ua.es)

**Abstract.** The South and South-eastern part of Spain exhibits the highest seismicity rate in the country. However, although the recently developed Quaternary Active Fault database of Iberia (QAFI, García-Mayordomo et al. (2012)) collected the available information existing in the study area regarding fault data for their use in seismic hazard applications, this information is of limited use since data quality is very heterogeneous: few earthquakes are associated to specific fault segments and occurrence time periods (when indicated) are affected by high uncertainties (Gaspar-Escribano et al., 2015). This fact has motivated the definition of alternative tectonic zonation models, to be used for evaluating the seismic hazard. So far, the clustering properties have not been considered in this regard, though they can provide essential information about the features of seismic energy release, depending on the tectonic style of a region (Talebi et al., 2024). This is why in this work the properties of the seismicity in terms of clustering are evaluated by applying the Nearest-Neighbor (NN) algorithm on the South-eastern Spain region. The scale parameters needed for the NN algorithm are optimised through the study of the z-score and the temporal anomalies between events in the identified clusters for each run. The tree structure under the graph theory notation has been proved useful in the determination of the critical threshold that separates the background (independent) seismicity from the clustered (dependent) seismicity in the NN algorithm. Once the clusters have been identified, the properties of the clusters have been quantified in terms of a selection of complexity measures: outdegree, closeness, and average node depth. This procedure has been applied by considering two different completeness magnitudes: Mw3.0 (the mean completeness magnitude for the entire catalogue) and Mw2.1 (accounting for the most recent part of the catalogue). The results are similar in terms of proportion of foreshocks, mainshocks and aftershocks, and indicate a clear distinction between the western-most part (higher complexity) and eastern-most part (lower complexity). To check this result, three different zonation models have been examined and cross-compared; two of them passed the Kolgomorov-Smirnov test, meaning the distributions of the selected complexity measures are not the same for the different zones defined in the models. These zonations can be used in order to assess the seismic hazard, as they account for the influence of the tectonic setting on the patterns of earthquakes occurrence, including the features of background and clustered seismicity components.

# 1 Introduction

South and South-eastern Spain are the areas within the Iberian Peninsula with the highest seismic hazard (IGN-UPM Working

Group, 2013; Kharazian et al., 2021). The tectonic setting in this region can be related to the geological features. In this sense, the main geological domains are the Betic Cordillera to the North, divided into Internal and External zones. The External Zone, divided into Prebetic and Subbetic, originally formed the south and south-east Mesozoic and Tertiary sedimentary cover of the Iberian shield and is arranged in many tectonic units (López-Casado et al., 2001). The main geological domains can be seen in Figure 1.

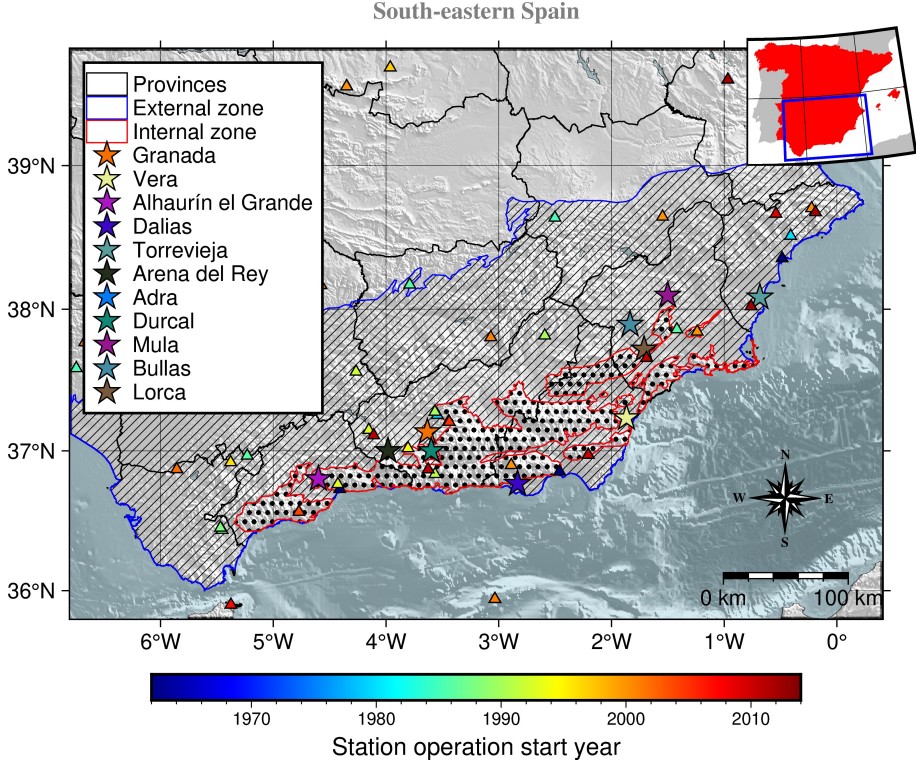

**Figure 1.** Main geological domains of South and South-eastern Spain (adapted from Buforn et al. (1995)). The red-edged dotted-filled polygons identify the Internal zone whereas the blue-edged polygons with a strip pattern fill mark the location of the External zone. The coloured stars represent the most damaging earthquakes in the area since the pre-instrumental era of the catalogue. The triangles represent the seismic stations' location, obtained from González, Á. (2017), and colour coded by the first year of operation.

South and South-eastern Spain experiences low-to-moderate seismic activity due to the collision of the Africa and Eurasia plates. Seismic energy is released mostly through small and occasional moderate earthquakes, typically at shallow depths, with a few rare deep events. The region's seismic history began with detection by local stations in the early 20th century, leading to the development of a national seismic network in the 1960s, with further improvements in the 1980s. Most earthquakes in the region can be classified as low magnitude, with the exception of notable events such as the 1910 Adra coast earthquake

(Mw6.2) and the deep 1954 Durcal earthquake (Mw7.0). Some of the most damaging earthquakes in the recent instrumental period have occurred in the Murcia region, i.e. Lorca's 2011 Mw5.1 earthquake, Mula's 1999 Mw4.9 and Bullas 2002 Mw5.0. All of them caused damage to the buildings and even the Lorca earthquake caused 9 deaths (Molina et al., 2018).

The historical seismicity of the region from the 15th to 20th centuries includes significant damaging earthquakes with onshore epicentres, such as those in 1431 (Granada), 1518 (Vera, Almería), 1680 (Alhaurín el Grande, Málaga) and 1804
(Dalias, Almería) with intensity VIII-IX (EMS-98) and estimated magnitude Mw >6.0 and the two most damaging earthquakes of our seismic catalogue: the 1829 Torrevieja earthquake and the 1884 Arena del Rey earthquakes, both with intensity IX-X (EMS-98) and estimated magnitude Mw >6.5 (Vidal-Sánchez, 1993).

The update of the Spanish seismic hazard map carried out in 2012 started with the identification of zones with different seismogenic characteristics. It is important to state that the high uncertainties in the QAFI database (García-Mayordomo et al.,
2012) and lack of earthquakes related to certain fault segments as pointed out by Gaspar-Escribano et al. (2015, p. 67) rules out using a fault based seismic source model. The ZESIS model (García-Mayordomo, 2015a; Gaspar-Escribano et al., 2015) originated from a previous one created following the expert judgment methodology after the cooperation of a large number of Earth Science researchers from Spain, Portugal, and France, in the frame of the first Iberian Meeting on Active Faults and Paleoseismology (Iberfault-2010), the European project SHARE (Seismic Hazard Harmonization in Europe) (García-
Mayordomo et al., 2010) and, eventually, by the Advisory Board for the New Seismic Hazard Map of Spain. The seismogenic source zones model can be consulted and downloaded from the *Instituto Geológico y Minero de España (IGME)* web under the name of ZESIS database (IGME, 2015): http://info.igme.es/zesis/. Although some of the tectonic characteristics are shared between all the subregions that were defined for the ZESIS zonation in South and South-eastern Spain, regions such as the Granada Basin are more prone to exhibit swarm-like seismic activity (Saccorotti et al., 2002; Stich et al., 2024). In this sense,
it is important to be able to identify the clustering characteristics of seismicity in different areas, as it could affect the seismic hazard analysis studies.

The declustering of the seismicity is one of the most important steps regarding the seismic hazard analysis, as one of the hypotheses is that the seismicity in the area follows a Poisson distribution (i.e., all the events are independent). This assumption cannot hold if the catalogue contains clustered seismicity data. Since the late XX century, different declustering algorithms
have been proposed: window methods, such as the Reasenberg-Jones' (Reasenberg, 1985; Reasenberg and Jones, 1989), the Gardner-Knopoff's (Gardner and Knopoff, 1974) or the Uhrhammer's (Uhrhammer, 1986); stochastic declustering methods (Zhuang et al., 2002; Zhuang, 2006) based on the Epidemic-Type Aftershock Sequence model (Ogata, 1998) are an example; correlation methods such as the Nearest-Neighbor algorithm (Zaliapin et al., 2008; Zaliapin and Ben-Zion, 2013a, 2020); etc. For a more detailed explanation of the declustering methods, we refer the reader to van Stiphout et al. (2012) work.
Performing a clustering analysis on the seismic catalogue has several benefits: 1) it enables working with a background seismicity catalogue (with independent events), 2) it enables the study of the time-dependent seismic hazard in seismic series, and 3) it could shed light on the mechanisms behind the seismic behaviour of certain areas by identifying the events in the clusters.

In this work, we apply the Nearest-Neighbor algorithm to the seismic catalogue of South and South-eastern Spain from 1970 up to the end of 2023. Our focus is to identify the main clusters present in the region. Then, we study the characteristics of the main clusters to see if there are important differences inside the region regarding the complexity and magnitude of the mainshocks. As a result of this analysis, a declustered catalogue will be obtained, which can be used for subsequent seismic hazard analysis.

## 2 Spanish Seismic Catalogue

Southern Spain is characterised by low-to-moderate shallow seismicity with rare high-magnitude earthquakes. The catalogue from the southern part of Spain (retrieved from https://www.ign.es/web/sis-catalogo-terremotos (IGN, 2022)) contains 46,296 earthquakes inside the region constrained by longitudes [7.0205ºW, 1.5526ºE] and latitudes [35.8762ºN, 39.8548ºN], from year 1970 until the end of 2023 and with depths shallower than 50 km. It has been homogenised following the equations from IGN-UPM Working Group (2013). The ranges of local magnitude for the application of such equations have been ignored for magnitude types 1, 2 and 4. The local magnitude or intensity scales can be checked at https://www.ign.es/web/resources/docs/IGNCnig/SIS-Tipo-Magnitud.pdf (IGN, 2025). A discussion on the homogenisation process can be read in the Appendix A. Only earthquakes belonging to tectonic zones in our study region with similar behaviour (crustal shortening direction) as defined in the ZESIS zonation (García-Mayordomo, 2015a) have been considered, as shown in Figure 2.

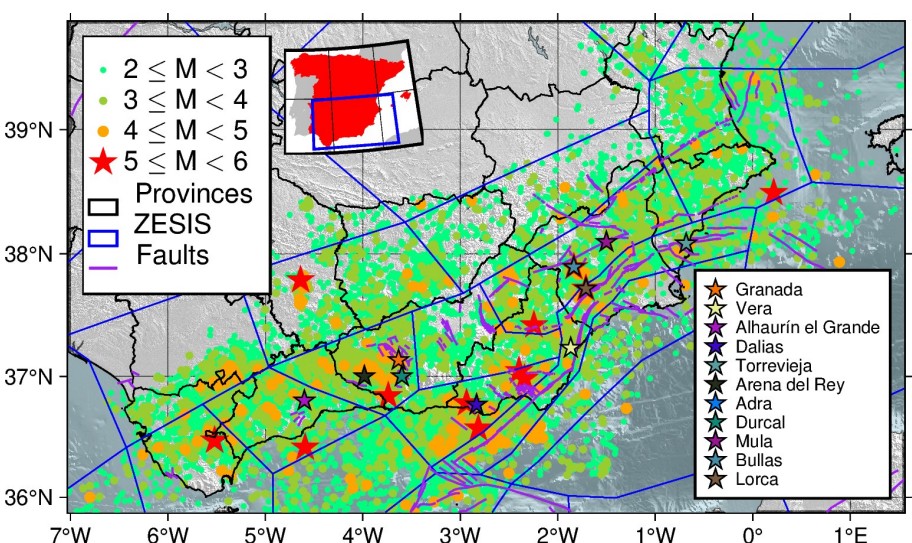

**Figure 2.** Catalogue of South and South-eastern Spain from 1970 to the end of 2023. It can be seen that faulting system determines the location of the epicentres for the most relevant earthquakes (Mw between 5.0 and 6.0 and marked as red stars) as most of them are located near these structures. The fault traces have been obtained from the QAFI database (García-Mayordomo et al., 2012; IGME, 2022) and the tectonic zonation polygons from the ZESIS database (IGME, 2015). The coloured stars represent the most damaging earthquakes in the area since the pre-instrumental era of the catalogue.

Figure 3 shows the depth-energy distribution along with the depth histogram as an inset. As can be seen, the seismicity is
concentrated around the 0 km and 10 km range, and decreases exponentially with depth. The magnitude ranges from 0.8 to
Mw5.4.

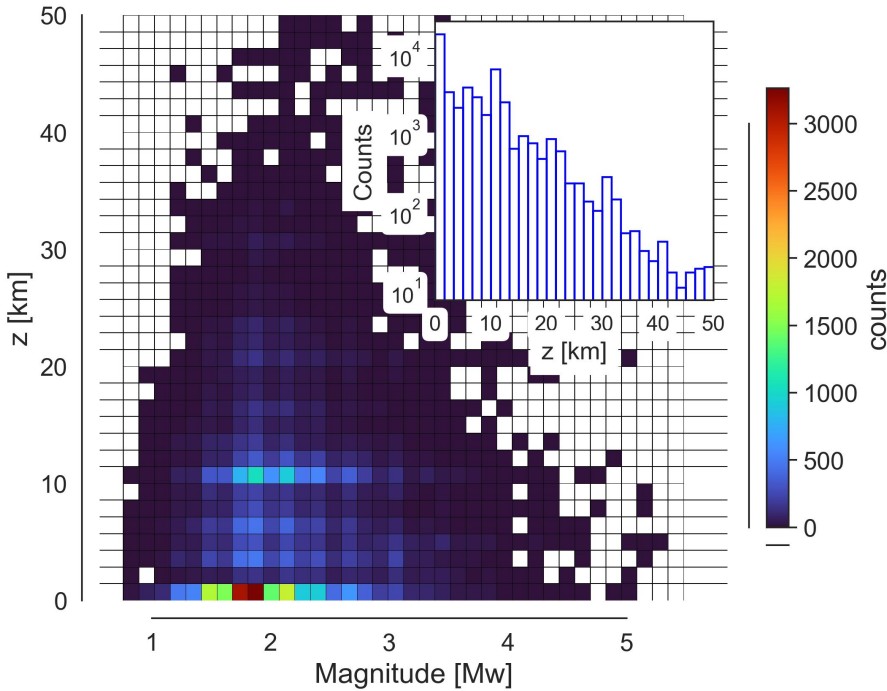

**Figure 3.** Magnitude-Depth distribution for the chosen catalogue. The histogram in the inset shows the depth distribution.

The detection sensitivity and, therefore, the completeness magnitude ($M_c$) of the catalogue have changed over time due to
upgrades in the seismic network. This fact is thoroughly discussed in the work of González, Á. (2017), the data from which
has made possible identifying four periods with distinct seismic network sensitivity: 1970-1984, 1985-1998, 1999-2013, and
2014-2023. These periods are also evident in Figure 5 by analysing the slope changes in the cumulative number of stations per
year. Figure 4 shows the events of the Spanish catalogue for the area of study. The number of events with magnitudes lower
than 3.0 increased from 1970-1984 to 1985 on, and then the number of events with magnitudes lower than 2.0 spiked from
1999 on, reflecting an improvement in the sensitivity of the Spanish seismic network.

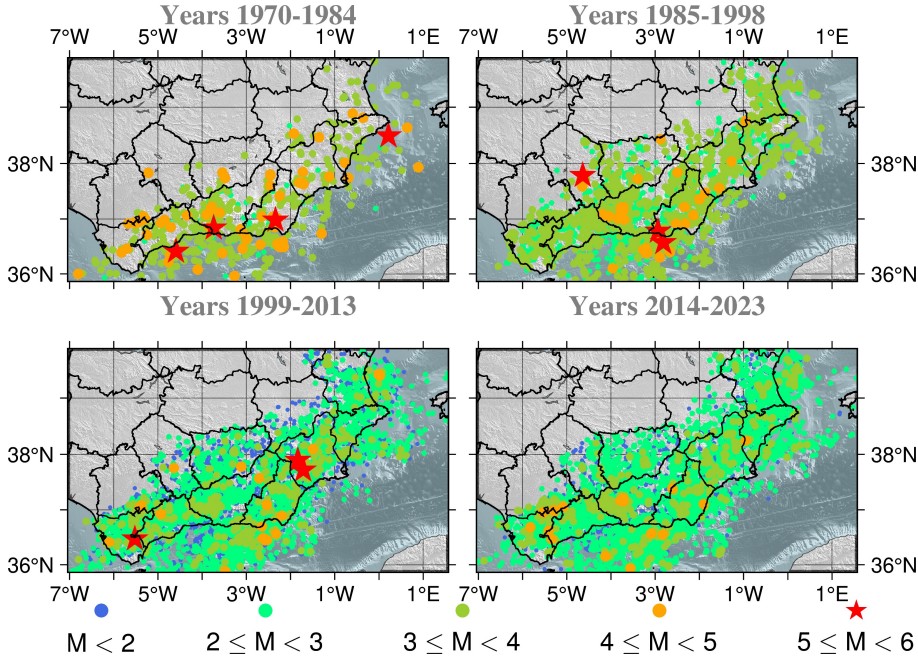

**Figure 4.** Spatial distribution of earthquake magnitudes in the four time intervals of the catalogue, identified as having different seismic network sensitivities: 1970-1984 (top left), 1985-1998 (top right), 1999-2013 (bottom left), and 2014-2023 (bottom right).

## 3  Nearest-Neighbor algorithm for clustering analysis

In this work, the Nearest-Neighbor algorithm (NN from now on) is applied to extract information about the clusters in the region of study. NN algorithm (Zaliapin et al., 2008; Zaliapin and Ben-Zion, 2013a, 2020), classifies the events in either background (those independent, typically assumed to follow a Poisson distribution) and clustered (those whose occurrence is related to other earthquakes, i.e., foreshocks and aftershocks). To do this, the spatiotemporal distances between each pair of events are computed following Baiesi and Paczuski (2004), using the times $t_k$ at which events occur, along with their hypocentre's coordinates $(\phi_k, \lambda_k, z_k)$ and magnitude $m_k$, for $k = 1, ..., N$ and $N$ the number of seismic events.

This distance between any pair of earthquakes $k$ and $j$ is defined as follows:

$$\eta_{kj} = \begin{cases} t_{kj} \, (r_{kj})^d \, 10^{-wm_k}, & t_{kj} \geq 0; \\ \infty, & t_{kj} < 0 \end{cases}, \tag{1}$$

where $t_{kj} = t_j - t_k$ are the times in-between events, $r_{kj}$ are the distances between the hypocentres or epicentres of the events, $d$ is the fractal dimension of the events in the catalogue, and $w$ is the parameter that inputs the weight of the magnitude in the spatiotemporal distance computation. The parameter $w$ usually equals the b-value of the Gutenberg-Richter (G-R) law (Gutenberg and Ritcher, 1944). The NN distance of each earthquake $j$ is then defined as $\eta_{ij} = \min_k \eta_{kj}$, thus identifying its

nearest neighbour $i$. The NN distance $\eta_{ij}$ can be further decomposed (Zaliapin et al., 2008) into the rescaled time, $T_{ij}$, and the rescaled space, $R_{ij}$, components (with $\eta_{ij} = T_{ij} \cdot R_{ij}$), which are defined as follows:

$$
\begin{cases}
T_{ij} = t_{ij} 10^{-qbm_i} \\
R_{ij} = r^d 10^{-pbm_i}, \quad \text{with } q + p = 1
\end{cases}
\tag{2}
$$

Importantly, Zaliapin and Ben-Zion (2013a) noticed that the NN distance exhibits a bimodal distribution that can be approximately decomposed into two Gaussian distributions: one corresponding to the background seismicity, and the other to the clustered seismicity. After calculating the threshold value $\eta_0$, which is typically the distance at which the probability densities of the two Gaussian distributions intersect, each event $j$ is classified as background seismicity if it is more than $\eta_0$ away from its nearest event $i$ (i.e., $\eta_{ij} > \eta_0$). Otherwise, it is classified as a clustered event and assigned a label identifying it as a foreshock,

aftershock, or mainshock, where the mainshock is defined as the largest magnitude event in the cluster, and foreshocks and aftershocks are the events that precede or follow it in time within the cluster. Each clustered event is also labelled by its *parent*, here represented by the mainshock. This information is needed to decluster the catalogue (keeping only the mainshocks and background events) and studying the cluster structure, by using the cluster data and the labels from the events along with their time and magnitude.

**3.1    Parameters for the NN algorithm**

The computation of the rescaled time and space of the NN distance requires both the b-value as defined in the G-R law and the fractal dimension of the events in the catalogue. Given that the seismic network's sensitivity in South-eastern Spain has changed over time, the b-value and completeness magnitude have been computed for each of the periods provided in Figure 4, as well as for the entire catalogue.

In Figure 5 the values obtained for each of the parameters are compared with the changes in the seismic network near the study area using the supplementary data provided by González, Á. (2017). The b-value and the completeness magnitude have been computed using ZMAP software (Wiemer, 2001). It can be seen that the completeness magnitude steadily decreases and stabilises, whereas the b-value increases sharply around 1980 when the seismic network starts its further development, and then decreases again to stay around the value obtained for the whole catalogue (b = 1.12).

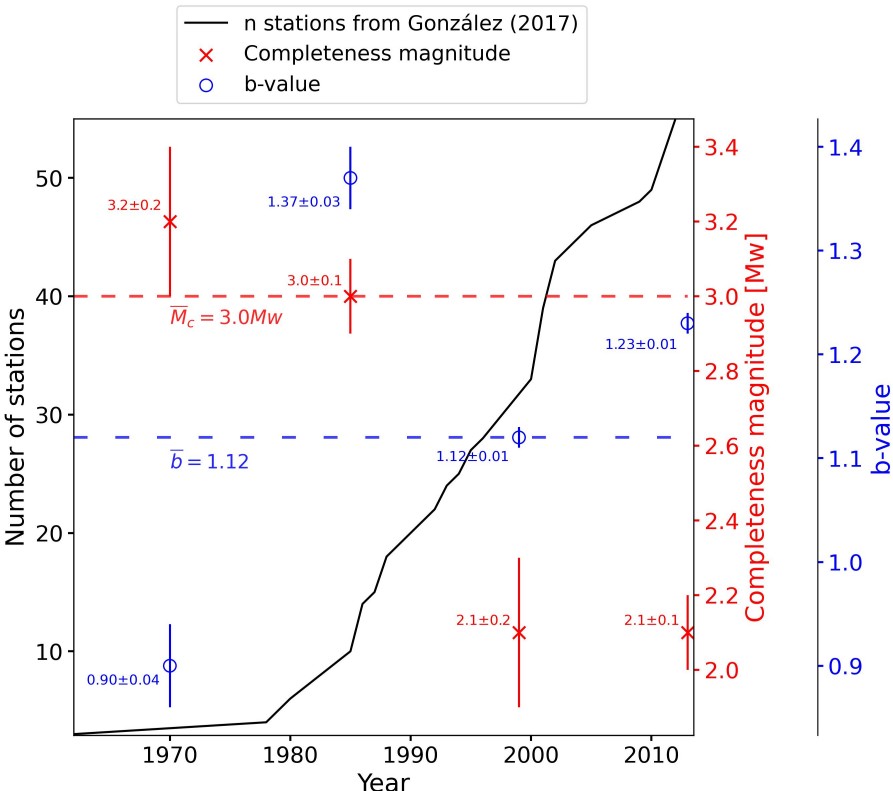

**Figure 5.** Changes in the b-value and completeness magnitude over time along with the evolution in the number of seismic stations near the area of study for the period 1970-2014. The blue circle markers represent the b-values and the red cross markers the completeness magnitude values from the year in which they are plotted to the next marker's year location. For example, the first blue marker would be the b-value from 1970 to 1984 (location of the next marker). The last marker would comprise the period from 2013 to 2023. The dashed lines indicate the values of the parameters for the whole catalogue (1970-2023).

Regarding the fractal dimension $d$, two different approaches have been used: 1) optimising the box-counting fractal dimension and 2) using the correlation integral approach as implemented in ZMAP.

To optimise the box-counting fractal dimension, two different parameters have been studied: the box size and the minimum number of events that a box must contain to be counted. The latter parameter has been restricted between a minimum number of one event and a maximum number given by the floor of $\sqrt{l}$, where $l$ is the longitude of the box in kilometres. This value has been selected by expert judgement.

Figure 6 shows the evolution of the fractal dimension when the minimum number of events per box and the size of the boxes are varied. Each iteration considers a higher value for the maximum number of events a box should contain to be considered in the computation of the fractal dimension.

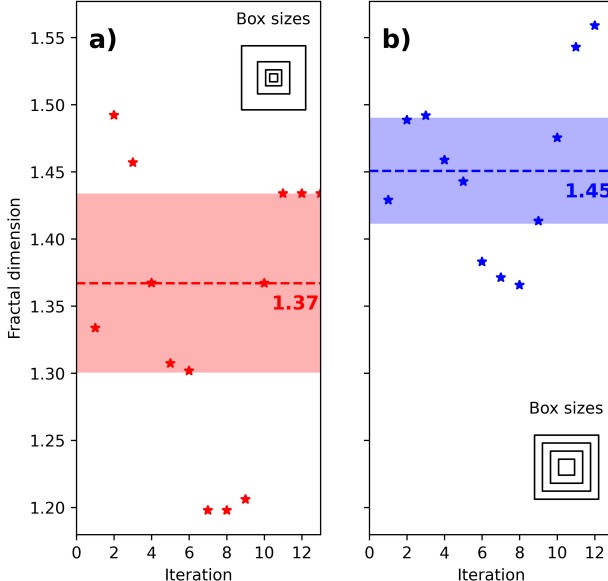

**Figure 6.** Fractal dimension computed with a) decreasing sizes using $l/2^n$, with $l$ the maximum box size and $n$ the step of reduction and b) decreasing by 10 km the size of the box from $l$ until 10 km. In this case $l$ has been set as 200 km. The dashed lines indicate the median value of the fractal dimension and the filled area around the median is one absolute median standard deviation.

Using a constant decreasing size of the box (Figure 6b) for the computation of the box-counting fractal dimension results in a more stable value for it, $\bar{d}_{b)} = 1.45 \pm 0.06$, when compared with the exponential decrease approach, $\bar{d}_{a)} = 1.37 \pm 0.09$.

The fractal dimension has also been obtained using the correlation integral method as implemented in the ZMAP software, obtaining a value of 1.5.

The influence on the variation of these parameters within the uncertainty limits has been studied in the Appendix B section, so two sets of parameters for the catalogue are obtained and shown in Table 1. These two sets of parameters define two subsets of earthquakes extracted from the catalogue, which we will refer to as the first and second datasets. The first dataset contains 1,806 events, and the second dataset, which includes the first, contains 20,057 events. Each dataset, equipped with its set of parameters, is analysed using the NN algorithm to associate each event $j$ with its nearest neighbour $i$ and NN distance $\eta_{ij}$.

**Table 1.** Optimal sets of parameters to be used in further cluster structure analysis.

| Parameter | First set | Second set |
|:---:|:---:|:---:|
| **b-value** | 1.0 | 1.0 |
| **Completeness magnitude, $M_c$ [Mw]** | 3.2 | 3.0 |
| **Fractal dimension, d** | 1.5 | 1.5 |

Then, for this two sets of parameters a further comparison on the anomalies due to the completeness magnitude can be seen in Appendix C.

Finally, it is important to point out that in order to compute the rescaled space component, $R_{ij}$, due to the large uncertainties in hypocentral depth determination, only epicentres have been considered. Following Zaliapin and Ben-Zion (2020) the analysis is performed considering epicentral coordinates, and thus is not affected by the depth uncertainties. When dealing with global scale analysis, where very deep earthquakes are reported, events are eventually selected within specified depth ranges (e.g. Zaliapin and Ben-Zion (2016)), while for narrow scale studies, where reliable depth information is available, it can be taken

into account (e.g.Martínez-Garzón et al. (2018)).

## 3.2    Role of $\eta_0$ in the identification of clusters

After associating each event with its nearest neighbour and calculating their NN distances, a threshold distance $\eta_0$ is chosen to separate background events from clustered events as well as to identify earthquake clusters.

Figure 7 shows the histogram of the NN distances obtained from the first dataset (panel a) and the second dataset (panel

b). As expected, the distribution of the NN distances is bimodal in both cases and can be approximated by a mixture of two Gaussian distributions, one relating to the background seismicity (yellow line) and the other to the clustered seismicity (orange line).

The threshold distance is set equal to the value in which the two estimated Gaussian distributions intersect: $\eta_0 = -3.4$ for the first dataset and $\eta_0 = -3.5$ for the second dataset (black solid vertical lines). After that, earthquake clusters are identified

in both datasets and those with the largest sizes are analysed to assess the choice of $\eta_0$ and gain a deeper understanding of its role in the declustering algorithm.

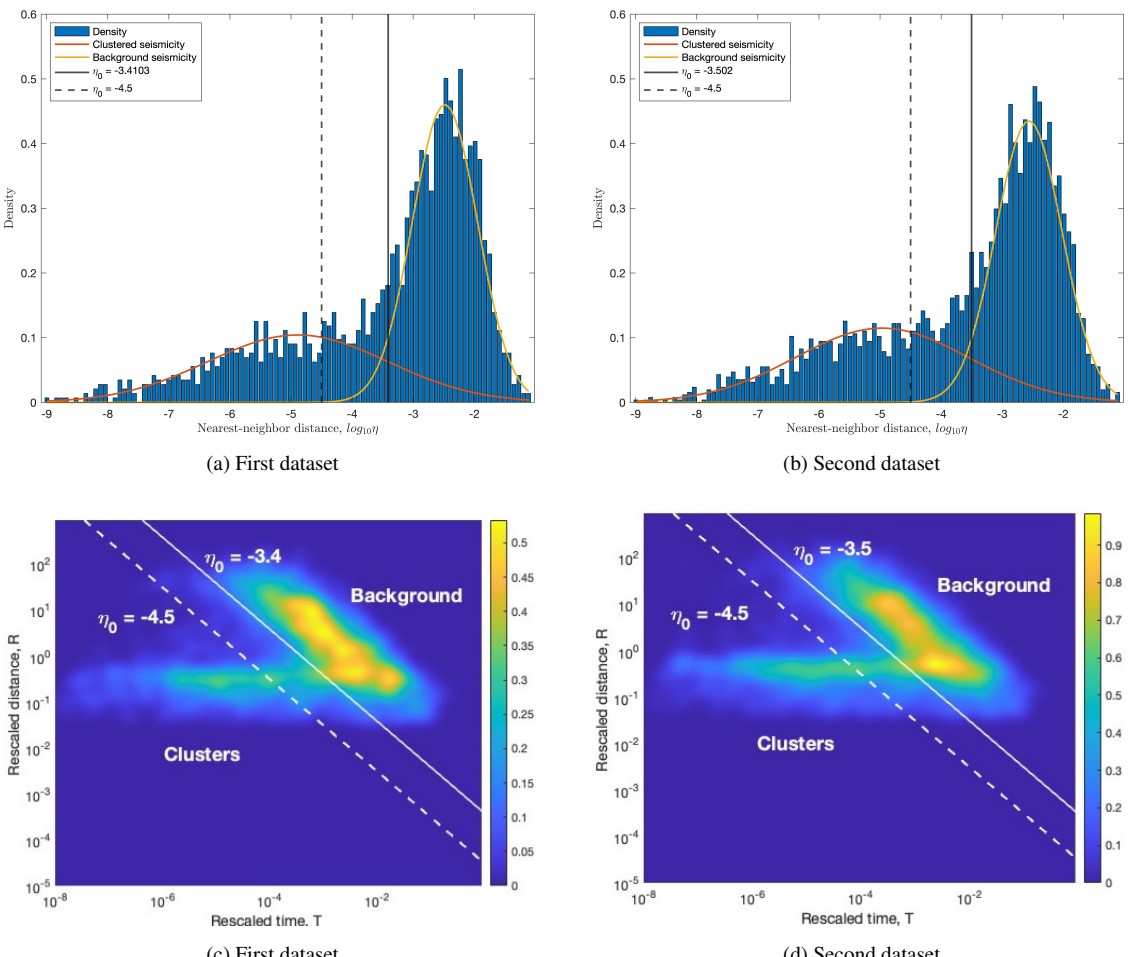

(a) First dataset

(b) Second dataset

(c) First dataset

(d) Second dataset

**Figure 7.** Histogram of the NN distances for (a) the first dataset and (b) the second dataset. Mixture model of two Gaussian density functions, one for background seismicity (yellow line) and the other for clustered seismicity (orange line), fitted to the datasets. Then, (c) and (d) show the 2D joint distribution of rescaled time and space for both datasets. The threshold distance $\eta_0$ given by the estimated values $-3.4$ in the panel (a) and (c) and $-3.5$ in panel (b) and (d) (solid vertical line), or given by the fixed value $-4.5$ (dashed vertical line in all the panels).

To illustrate the analysis we performed on the most populated clusters, we consider two large clusters identified in both datasets: the Adra's sequence, whose mainshock occurred on December 23, 1993, Mw5.2; and the Granada's swarm, whose mainshock occurred on January 23, 2021, Mw4.4. Figures 8a and 8c show the cluster structures of the Adra's sequence identified in the first dataset with $\eta_0 = -3.4$ and in the second dataset with $\eta_0 = -3.5$, respectively. In Figures 9a and 9c, the spatial distributions of the epicentres are also shown for these clusters. Similarly, the results of the Granada's swarm identified in both datasets by using the estimated values of $\eta_0$ are shown in Figures 10a, 10c and Figures 11a, 11c.

An obvious note is that, for both seismic sequences, the events are more numerous in the cluster of the second dataset than in that of the first dataset (the lower the completeness magnitude, the more events in the cluster).

In Figure 8a the mainshock of the cluster is preceded by a foreshock. A closer inspection in the spatial distribution of magnitude reveals that this foreshock must be an anomaly, as there is more than 1° difference in longitude between foreshock and the mainshock (Figure 9a). The same argument holds for those aftershocks that directly follow the mainshock, but which in turn generate few aftershocks.

It is noted that even the Granada's swarm clusters identified in the two datasets have events anomalously distant from the mainshock (Figures 10a and 10c).

To prevent the erroneous classification of distant events into clusters, which are likely background events, we manually adjusted the estimated distance threshold values for the two datasets by gradually lowering them until a satisfactory fixed value of $\eta_0 = -4.5$. Appendix D shows the difference in the anomaly count in the clusters when fixing a lower $\eta_0$ value.

The right panels in Figures 8-9 show the resulting clusters of the Adra's sequence obtained from the two datasets with tuned parameter $\eta_0 = -4.5$. Similarly, the right panels in Figures 10-11 show the resulting clusters of the Granada's swarm.

From the comparison of the pairwise panels on the left and right in Figures 8-11 (the clusters comes from the same dataset, but with different $\eta_0$ values), we note that the complexity in the cluster structures slightly changes by using the estimated ($-3.4$, $-3.5$) or the tuned ($-4.5$) $\eta_0$ values. Nevertheless, the choice of $\eta_0$ influences the spatial dispersion of the events belonging to the cluster: events that are anomalously distant from their mainshocks are excluded from the clusters when using $\eta_0 = -4.5$.

As a result of this study, we will focus only on the clustering results obtained by the following parameters: (1) the completeness magnitude $M_c = 3.2$ (first dataset), since only minor changes in the cluster structures are observed when it is lowered to $M_c = 3.0$; (2) the tuned value $\eta_0 = -4.5$, as it more effectively defines the cluster structure (eliminating anomalous foreshock activity and/or distant events) and prevents background events from being misclassified as clustered events, which could significantly affect the cluster characterisation. In the following figures, the fault names have been shortened in order for them not to hinder the visualization of the plots. The complete names according to QAFI 4.0 database (García-Mayordomo et al., 2012; IGME, 2022) are shown in the following table:

| A: Adra | AF: Alfacar | AJ: Alitaje |
|---|---|---|
| ALB: Albuñuelas | AT: Atarfe | B-A: Belicena-Alhendín |
| BA: Balanegra | BZ: Baza | CA: Carboneras |
| CU: Cubillas | D: Dílar | E: Escóznar |
| EGRFS: Eastern Gador Range Fault System | F-J: El Fargue-Jun | G-L FZ: Graena-Lugros Fault Zone |
| GR: Granada | GU: East of Guadix | H: Huenes |
| LdV: Loma del Viento | O-PP: Obéilar - Pinos Puente | P-N: Padul-Nigüelas |
| PP: Pinos Puente | PR: Pedro Ruiz | SdZFZ: Solana de Zamborino Fault Zone |
| SF: Santa Fe | T-O: Tocón-Obéilar | VdZ: Ventas de Zafarraya |

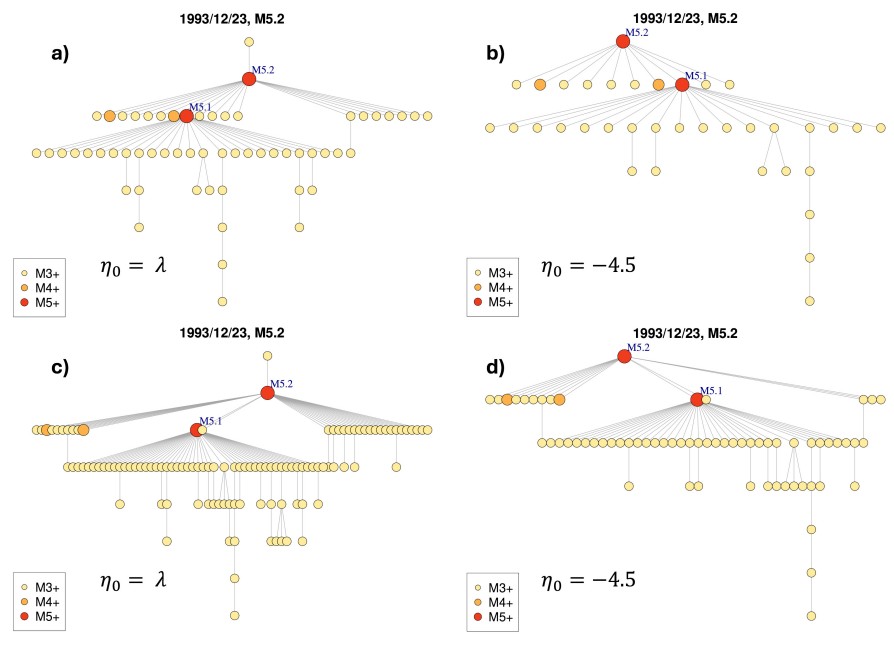

**Figure 8.** Adra's sequence, whose mainshock occurred in December 23, 1993, Mw5.2. Cluster structures obtained from the different set of parameters and critical threshold values: a) first dataset, estimated $\eta_0 = -3.4$; b) first dataset, tuned $\eta_0 = -4.5$; c) second dataset, estimated $\eta_0 = -3.5$; d) second dataset, tuned $\eta_0 = -4.5$.

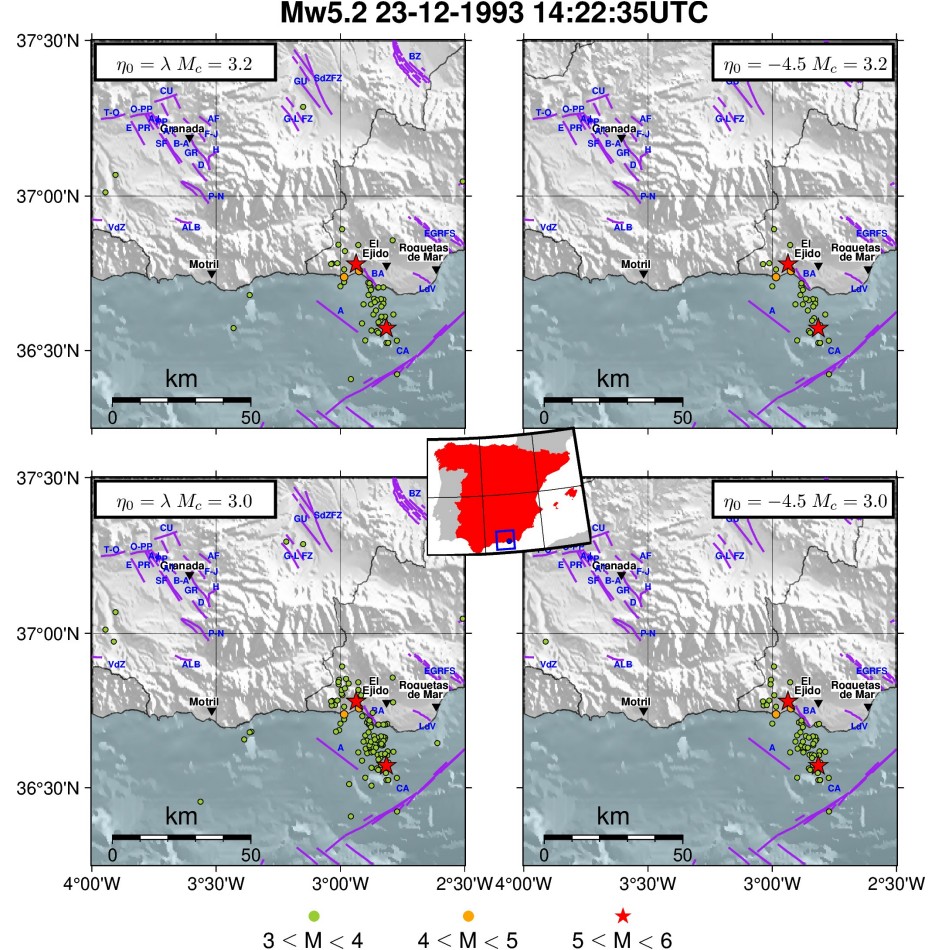

**Figure 9.** Adra's sequence, whose mainshock occurred in December 23, 1993, Mw5.2. Spatial distributions of the earthquakes' epicentres in the clusters obtained from the different set of parameters and critical threshold values: a) first dataset, estimated $\eta_0 = -3.4$; b) first dataset, tuned $\eta_0 = -4.5$; c) second dataset, estimated $\eta_0 = -3.5$; d) second dataset, tuned $\eta_0 = -4.5$. The central inset shows the location of the cluster's mainshock in Spain and the purple lines in each of the subplots mark the position of the active faults from QAFI v4.0 (García-Mayordomo et al., 2012; IGME, 2022) in the area. The blue labels indicate the different confirmed fault names and the black labels with the inverted triangle mark the position of the main cities in the area (population greater than 30,000).

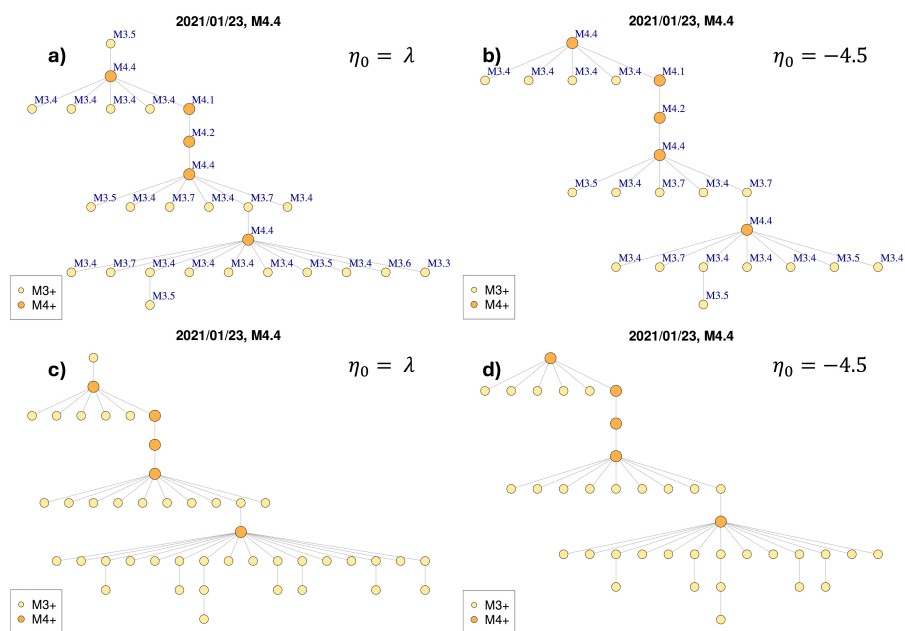

**Figure 10.** Granada's swarm, whose mainshock occurred in January 23, 2021, Mw4.4. Cluster structures obtained from the different set of parameters and critical threshold values: a) first dataset, estimated $\eta_0 = -3.4$; b) first dataset, tuned $\eta_0 = -4.5$; c) second dataset, estimated $\eta_0 = -3.5$; d) second dataset, tuned $\eta_0 = -4.5$.

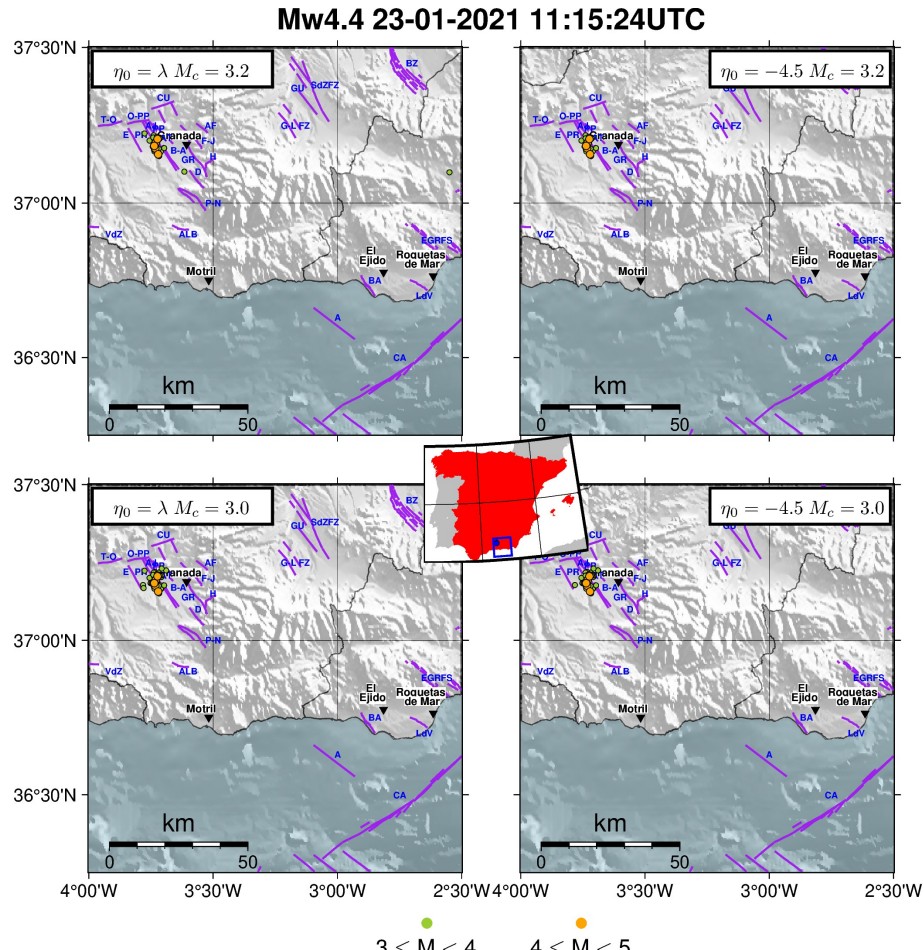

**Figure 11.** Granada's swarm, whose mainshock occurred in January 23, 2021, Mw4.4. Spatial distributions of the earthquake's epicentres in the clusters obtained from the different set of parameters and critical threshold values: a) first dataset, estimated $\eta_0 = -3.4$; b) first dataset, tuned $\eta_0 = -4.5$; c) second dataset, estimated $\eta_0 = -3.5$; d) second dataset, tuned $\eta_0 = -4.5$. The central inset shows the location of the cluster's mainshock in Spain and the purple lines in each of the subplots mark the position of the active faults from QAFI v4.0 (García-Mayordomo et al., 2012; IGME, 2022) in the area. The blue labels indicate the different confirmed fault names and the black labels with the inverted triangle mark the position of the main cities in the area (population greater than 30,000).

## 4 Graph analysis of earthquake clusters

An earthquake cluster identified by the NN algorithm can be structured as a tree graph, illustrating the links between events and their nearest neighbours.

Clusters can be studied from a graph-theory point of view to identify their features and assess the complexity of their structures. Based on graph analysis, Zaliapin and Ben-Zion (2013b) found two distinct types of cluster sequences, referred

as burst-like (with an umbrella-like shape) and swarm-like (with a chain-like tree graph shape) sequences, whose spatial variability helps characterize Californian regions with different heat flow and other viscosity-related properties. Similarly, Varini et al. (2020) used two other scalar measures of tree graphs to describe the structure of the clusters and classify them according to their complexity in North-eastern Italy. Some of these scalar measures will be used in this study to assess the complexity of clusters containing 5 or more events and to examine spatial patterns in the study area.

## 4.1 Scalar measures of tree graphs

Let the tree graph representation of an earthquake cluster be denoted by $T$, where the nodes of $T$ are the events within the cluster, and the edges are set between the nodes and their nearest neighbours. Let $|T|$ be the number of nodes in $T$, i.e. the cluster's size.

### Outdegree centralisation

The number of outgoing edges from a node $v$ is named outdegree of $v$ and denoted by $\delta(v \mid T)$. The *outdegree centrality* of $v$ is then defined as $c_\delta(v \mid T) = \delta(v \mid T)/(|T|-1)$, that is the ratio between the outdegree of $v$ and the maximum possible outdegree $|T|-1$ of a node in a general tree of size $|T|$. Outdegree centrality of $v$ takes values in $[0,1]$. The higher the outdegree centrality of $v$, the more important the node is, meaning that $v$ is more central in the tree $T$ as it has more outgoing edges than other nodes. Given the node $v^*$ with highest outdegree centrality in the tree $T$, the *outdegree centralisation* of $T$ is given by:

$$C_\delta(T) = \frac{\sum_{v \in T}[c_\delta(v^* \mid T) - c_\delta(v \mid T)]}{|T|-1} \,, \tag{3}$$

which represents the difference between the outdegree centrality of the node with highest centrality and that of all other nodes, normalised with respect to the maximum possible analogous difference in a general tree of size $|T|$. The outdegree centralisation is also an index in $[0,1]$ that summarises the outdegree centralities of all nodes in $T$. A high outdegree centralisation of $T$ indicates the presence of nodes with high outdegree centralities. For example, burst-like clusters have outdegree centralisations close to 1.

### Closeness centralisation

The *closeness centrality* $c_c(v|T)$, as defined by Bavelas (1950), gives an idea of how close a node $v$ is to the rest of the nodes $w$ in the tree $T$. It can be calculated as the reciprocal of the ratio between the sum of the lengths $d(v,w)$, and the minimum possible analogous sum in a general tree of size $|T|$: $c_c(v|T) = (|T|-1)/\sum_{w \in T} d(v,w)$. Closeness centrality of node $v$ also ranges in $[0,1]$. The higher the closeness centrality of $v$, the more important the node is, meaning that $v$ is more central in the tree $T$ because it is well connected to the other nodes by paths. Given the node $v^*$ with highest closeness centrality in the tree $T$, the *closeness centralisation* of $T$ is given by:

$$C_c(T) = \frac{\sum_{v \in T}[c_c(v^* \mid T) - c_c(v \mid T)]}{(|T|-1)^2/|T|} \,, \tag{4}$$

which represents the difference between the closeness centrality of the node with highest centrality and that of all other nodes, normalized with respect to the maximum possible analogous difference in a general tree of size $|T|$. The closeness centralisation

also varies in $[0, 1]$. A high closeness centralisation of $T$ indicates the presence of nodes that are well connected to the others. For example, burst-like clusters have closeness centralisations close to 1.

**Average leaf depth**

The average leaf depth $\langle d \rangle$ was introduced by Zaliapin and Ben-Zion (2013b) and is evaluated by considering some special nodes in $T$: the root and the leaves. The root corresponds to the first event in time within the cluster, while the leaves represent events that are not nearest neighbours of other nodes. In other words, the root is linked to all other nodes by a path and the leaves have a null outdegree. Given a leaf of $T$, its *leaf depth* is the length of the shortest path between the root and this leaf. Therefore, the *average leaf depth* $\langle d \rangle$ is given by the sum of all leaf depths divided by the number of leaves in $T$. In general, clusters with an burst-like structure have low average leaf depth values (closer to 1), as most of the nodes sprout from the root of the tree; chain-like clusters, on the other hand, have high average leaf depth values (closer to $|T| - 1$).

In summary, values close to 1 for both centralisation measures indicate simple burst-like cluster structures, referred to as burst-like sequences by Zaliapin and Ben-Zion (2013b), because the mainshock node is directly connected to most of the aftershock nodes. Smaller values of the centralisation measures correspond to more complex chain-like/swarm-like clusters. On the contrary, small (large) average leaf depth values denote simple burst-like (complex swarm-like) clusters. However, it should be noted that this last measure is not normalised (it does not have a constant upper bound) and, unlike the centralisation measures, is influenced by the cluster size.

Figure 12 shows an example of such cluster structure types according to their tree graphs. In reality more complex cluster structures are expected due to combination of these typologies. For instance, Adra's sequence (Figure 8) could be classified as a double-burst-like cluster whereas Granada's swarm (Figure 10) has a more pronounced chain-like component.

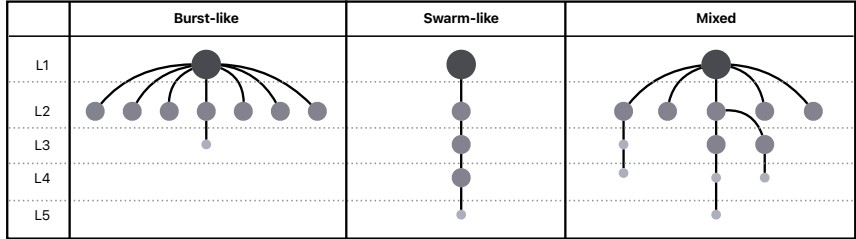

**Figure 12.** Cluster types according to their tree graph structures. The events are represented with sizes and grey-shade colours according to their magnitude.

## 4.2 Results for completeness magnitude Mw3.2

Based on previous results, earthquake clusters containing five or more events are considered, as obtained from the NN algorithm with a tuned threshold distance $\eta_0 = -4.5$ applied to the South-eastern Spain catalogue with completeness magnitude Mw3.2.

Figure 13 shows four maps where the mainshock epicentres of these clusters are coloured according to the values of the out-degree centralisation (top left), closeness centralisation (top right), and average leaf depth (bottom left) for their corresponding clusters, as well as to the mainshock magnitudes (bottom right). The yellow colour corresponds to simple burst-like clusters, whereas clusters become more complex with swarm-like features as the colour shifts towards dark red. Based on the outdegree and closeness centralisation maps, the south-western part of the region is dominated by more swarm-like clusters (e.g., the

Mw4.4 Granada swarm) than elsewhere. By comparing the centralisation maps with the magnitude map, we observe that the north-eastern part of the region was affected by the strongest earthquakes that occurred during the study period (1970-2023) and their centralisation values mostly approach simple burst-like behaviours (e.g., the Mw5.2 Adra sequence). This aligns with the behaviour commonly associated with strong earthquakes, which are known to generate numerous aftershocks. All these remarks are also consistent with the results in the average leaf depth map, which closely resemble those of closeness

centralisation. This was expected, as both measures are based on selected path lengths between nodes.

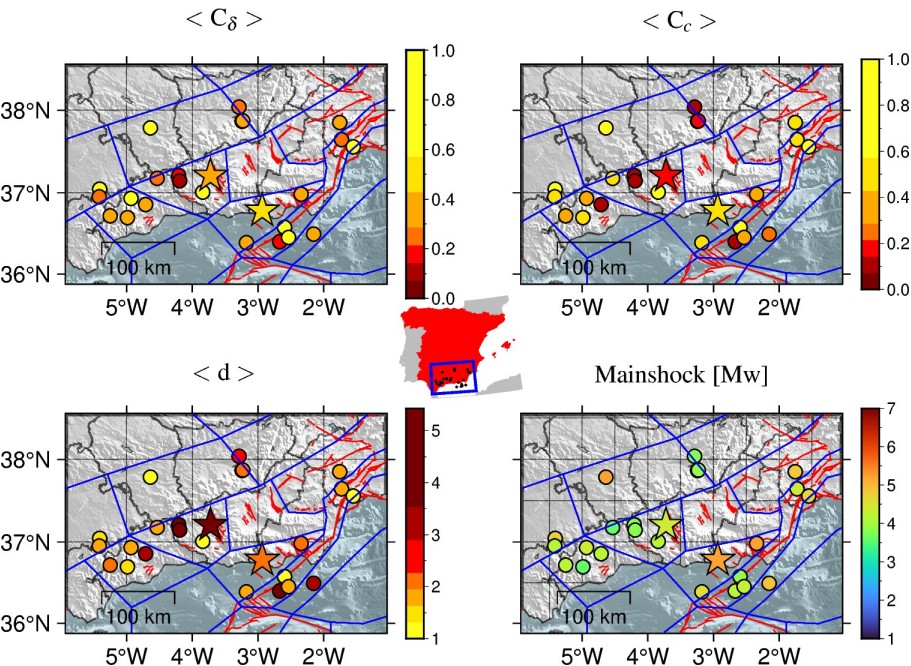

**Figure 13.** Maps of the mainshock epicentres of the clusters identified by NN algorithm with $\eta_0 = -4.5$ and completeness magnitude Mw3.2. Colours represent the cluster values of the (top left) outdegree centralisation, (top right) closeness centralisation, average leaf depth (bottom left), and mainshock magnitude (bottom right). The mainshocks of both Adra's sequence (north-west) and Granada's swarm (south-east) have been marked with stars. The red lines mark the position of the active faults from QAFI v4.0 (García-Mayordomo et al., 2012; IGME, 2022) in the area and the blue lines the tectonic zonation for Spain (García-Mayordomo, 2015b; IGME, 2015). The centre inset shows the location of the epicentres within Spain.

## 4.3 Results for completeness magnitude Mw2.1

The same analysis was conducted by lowering the completeness magnitude to Mw2.1 and shortening the study period from 1999 (instead of 1970) to 2023, according to the results on the completeness of the catalogue in Figure 5. The motivation lies in verifying whether the results obtained so far remain valid in a complete catalogue that includes low-magnitude events, even if the time period is necessarily shorter.

Table 2 compares the NN declustering of the two catalogues with completeness thresholds, Mw3.2 and Mw2.1, respectively, presenting the total number of events and those classified as background activity (singles) and clustered activity. The clustered events are further classified as foreshocks, mainshocks, and aftershocks. On the one hand, the smaller size of the previously studied dataset with $M_c$ 3.2 made it easier to manage and visually display the clusters. On the other hand, the new dataset with $M_c$ 2.1 offers the advantage of including a significantly larger number of events, including those of lower magnitude. It can be seen that the clustered to background seismicity ratio increases as the completeness magnitude decreases. This is to be expected, as lower magnitude earthquakes are more frequently associated with seismic sequences in the aftershock category. It is also in agreement with the increase in the ratio of aftershocks (17% for the Mw3.2 completeness magnitude vs 25% for the Mw2.1 completeness magnitude).

**Table 2.** Summary of the clustering statistics for the two different completeness magnitudes.

| | | | | Full catalogue | | |
|---|---|---|---|---|---|---|
| Mc | n events | CtoB ratio | Singles | Clusters | | |
| | | | | Foreshocks | Mainshocks | Aftershocks |
| 3.2 | 1806 | 0.43 | 1,262 (70%) | 92 (5%) | 149 (8%) | 303 (17%) |
| 2.1 | 20,057 | 0.64 | 12,210 (61%) | 1,244 (6%) | 1,643 (8%) | 4,960 (25%) |

**CtoB**: Clustered seismicity to Background seismicity ratio.

The maps in Figure 14 illustrate the results of the new additional analysis. The comparison with Figure 13 shows that the new findings are consistent with previous results: the mainshocks in the south-western part of the region exhibit low-to-moderate magnitude and their associated scalar measures suggest swarm-like behaviour. In contrast, the north-eastern part of the region has experienced recent strong earthquakes, with the scalar measures indicating burst-like activity.

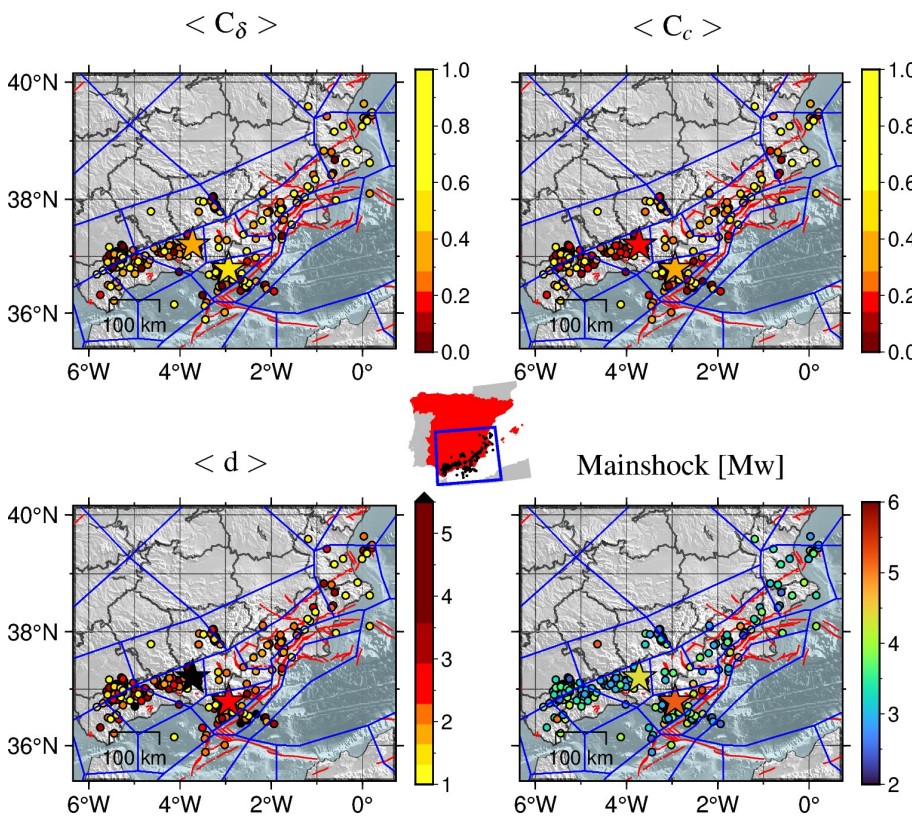

**Figure 14.** Maps of the mainshock epicentres of the clusters identified by NN algorithm with $\eta_0 = -4.5$ and completeness magnitude Mw2.1. Colours represent the cluster values of the (top left) outdegree centralization, (top right) closeness centralization, average leaf depth (bottom left), and mainshock magnitude (bottom right). The mainshocks of both Adra's sequence (north-west) and Granada's swarm (south-east) have been marked with stars. The red lines mark the position of the active faults from QAFI v4.0 (García-Mayordomo et al., 2012) in the area and the blue lines the tectonic zonation for Spain (García-Mayordomo, 2015b; IGME, 2015). The centre inset shows the location of the epicentres within Spain.

## 5 Probing seismic zonation through cluster analysis

It is interesting to compare then the results for both completeness magnitude thresholds in terms of average node depth and cluster size (Figure 15), as this representation can provide some insights on the possible relation between the cluster size and their different types. It can be observed that, especially for Mc=2.1, the average node depth <d> naturally increases with the cluster's size for swarm-like sequences, while for burst-like sequences <d> remains quite low even for large clusters, composed by more than a hundred events. Moreover, considering the geographical information (longitude values) it can be seen that in

the west zone (longitude < -3°) complex clusters (higher average node depth) are more common than in the east zone, even for similar cluster sizes.

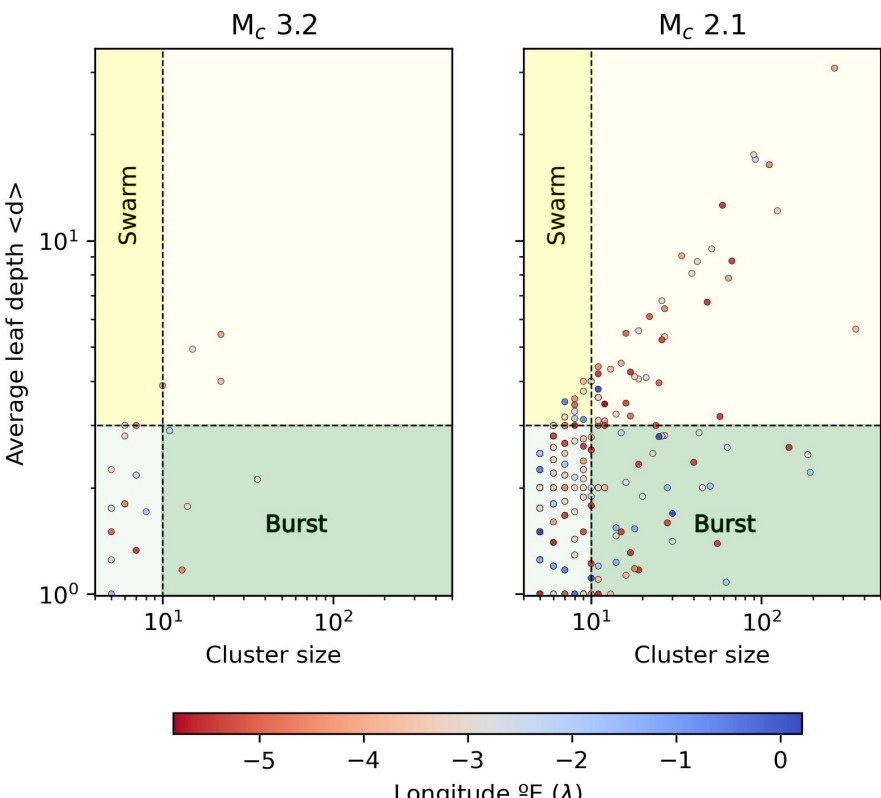

**Figure 15.** Average leaf depth vs cluster size for completeness magnitude 3.2 (left) and completeness magnitude 2.1 (right). The red markers are events at the west side of the area of study (longitude < -3º) and those blue-coloured belong to the east side (longitude > -3º).

This motivates us to proceed with the clustering analysis of the dataset with completeness magnitude Mw2.1 in order to propose possible zonation models for the study area complementing the information provided by the ZESIS tectonic zonation (García-Mayordomo, 2015a). In Figure 16, the study area was divided into square cells of 0.5º in length, with only those cells containing at least 4 mainshocks being considered.

Using the information from Figure 14 the median values of the scalar measures within the cells are calculated, along with the median magnitudes of the mainshocks. The resulting median values are then represented by colours in the cells according to a specified colour scale (Figure 16); the background areas represent the external zones (blue area) and internal zones (red area) as defined in Figure 1.

All maps in Figure 16 shows a clear separation between two different regions: the western region is characterized by complex swarm-like clusters with mainshocks of low to moderate magnitude (dominance of red-orange cells in the scalar measure maps and of blue-light blue cells in the magnitude map), while the eastern region displays simple burst-like clusters with even stronger mainshocks (dominance of yellow-light orange cells in the scalar measure maps and of green cells in the magnitude map).

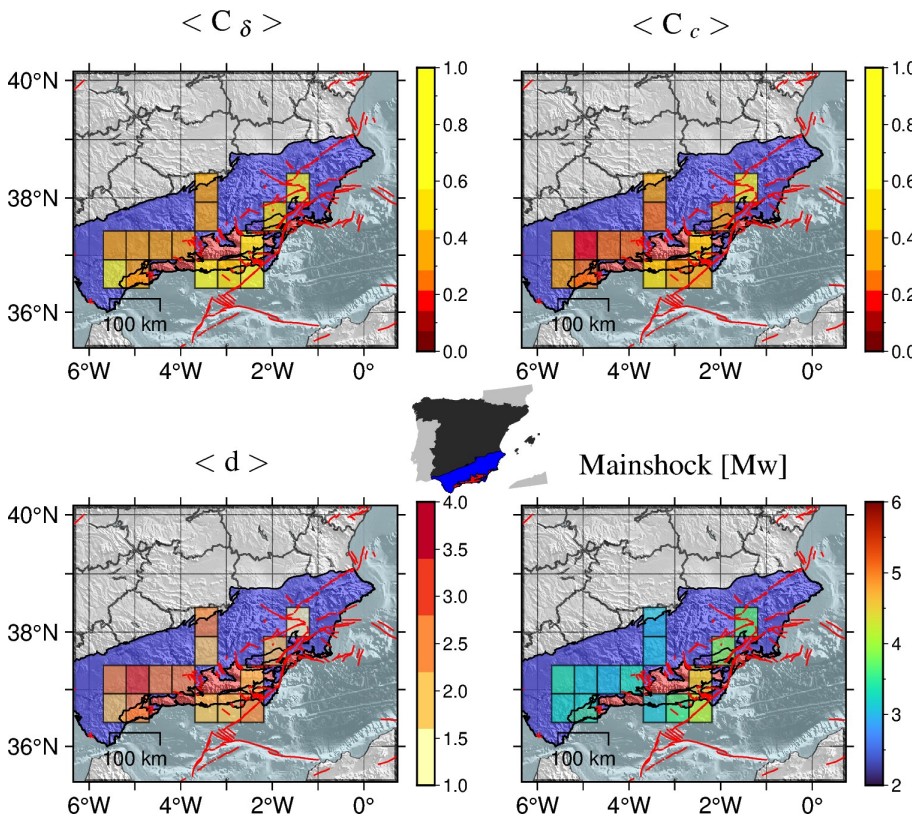

**Figure 16.** Gridded median values of the outdegree centralization (top left), closeness centralization (top right), average leaf depth (bottom left), and mainshock magnitude (bottom right). The background areas represent the external zones (blue area) and internal zones (red area) as defined in Figure 1. The red lines mark the position of the active faults from QAFI v4.0 (García-Mayordomo et al., 2012; IGME, 2022) in the region.

Based on the spatial distribution of the scalar measures under consideration (Figure 16) and on the geographic map of the geological domains (Figure 2), we propose three zonation models, using the ZESIS tectonic zones as building pieces. These zonations are illustrated in Figure 16. It can be seen that there is a slight difference between Zonation 1 and Zonation 2, as Zonation 2 includes an additional ZESIS tectonic zone. Zonation 3 is a modified version of Zonation 2, whose zone 1 is further divided into an east zone and a west zone. This has been done in order to check if further subdivisions of the most

populated zone (in terms of seismic events) show different characteristics in the regional scale for the studied parameters. If such variations exist, a more complex zonation model could be proposed.

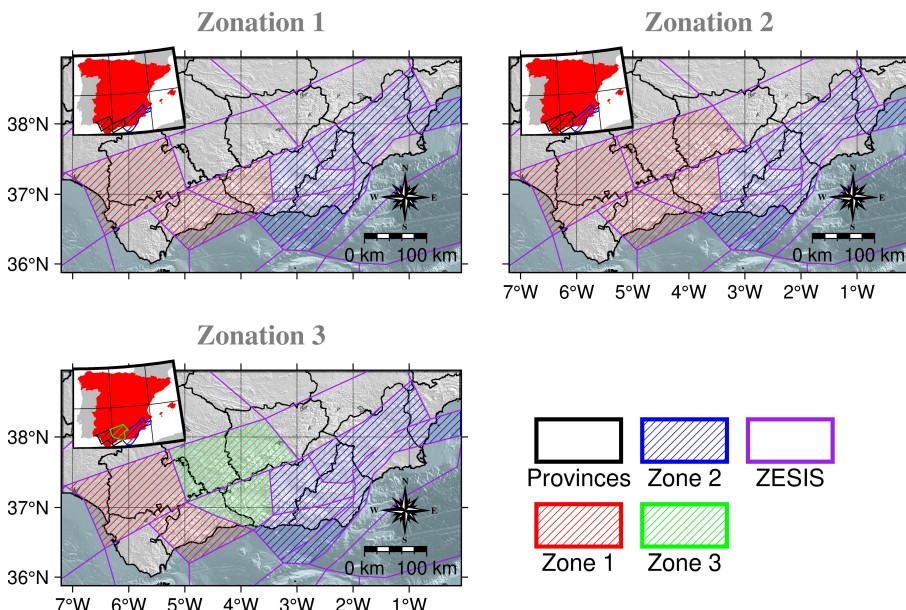

**Figure 17.** The three proposed zonation models of the study area based on the identified clustering features and the ZESIS tectonic zonation (García-Mayordomo, 2015b; IGME, 2015).

To assess the consistency of the proposed zonation models with the assumptions made, we investigate whether the values of the analysed scalar measures (i.e., outdegree centralizations, closeness centralization, and average leaf depth) differ significantly across the zones of a zonation. The two-sample Kolgomorov-Smirnov (KS) test (Rohatgi and Ehsanes Saleh, 2015) is used to verify the null hypothesis that two samples originate from the same population, versus the alternative hypothesis that they come from different populations. The results of KS tests for Zonation 1 are summarised in Table 3. Since the p-value is below the significance level 0.05 for all scalar measures, the null hypotheses are rejected, indicating that the distributions of the scalar measures are significantly different between zones 1 and 2. The KS tests applied to Zonation 2 provide similar results (Table 4).

Similarly, Table 5 shows the comparison between pairs of the three zones for Zonation 3. The analysis confirms that zones 1 and 2 display significantly different distributions in relation to the scalar measures under consideration. The same holds true for zones 2 and 3. In contrast, zones 1 and 3 do not exhibit such differences; this suggests that these two zones of Zonation 3 might be better combined into one, as in Zonation 2, rather than kept separate. In conclusion, Zonation 2 is preferred due to its broader coverage of the study region compared to Zonation 1, although Zonation 1 has proven equally valid in this analysis.

**Table 3.** Zonation 1: Output of the KS tests comparing the distributions of scalar measure values in its zones 1 and 2, whose sample size is 115 and 67 respectively.

| | | | | Zone 1 | Zone 2 |
| --- | --- | --- | --- | --- | --- |
| **Parameter** | **Max. Neg** | **Max. Pos.** | **p-value** | **Mean (StDev)** | **Mean (StDev)** |
| **Outdegree** | $-0.242$ | 0.000 | $< .025$ | 0.446 (0.273) | 0.563 (0.283) |
| **Closeness** | $-0.331$ | 0.006 | $< .001$ | 0.370 (0.286) | 0.477 (0.272) |
| **Avg. Node Depth** | $-0.006$ | 0.296 | $< .005$ | 3.073 (3.413) | 2.239 (2.089) |

**Table 4.** Zonation 2: Output of the KS tests comparing the distributions of scalar measure values in its zones 1 and 2, whose sample size is 155 and 67 respectively.

| | | | | Zone 1 | Zone 2 |
| --- | --- | --- | --- | --- | --- |
| **Parameter** | **Max. Neg** | **Max. Pos.** | **p-value** | **Mean (StDev)** | **Mean (StDev)** |
| **Outdegree** | 0.000 | 0.226 | $< .025$ | 0.454 (0.272) | 0.563 (0.283) |
| **Closeness** | $-0.008$ | 0.318 | $< .001$ | 0.370 (0.280) | 0.477 (0.272) |
| **Avg. Node Depth** | $-0.260$ | 0.008 | $< .005$ | 2.972 (3.090) | 2.239 (2.089) |

**Table 5.** Zonation 3: Output of the KS tests comparing the distributions of scalar measure values for all pairwise combinations of its zones 1, 2 and 3, with sample sizes 72, 67, and 83, respectively.

| | | | | Zone 1 | Zone 2 |
|---|---|---|---|---|---|
| **Parameter** | **Max. Neg** | **Max. Pos.** | **p-value** | **Mean (StDev)** | **Mean (StDev)** |
| **Outdegree** | 0.000 | 0.199 | > .100 | 0.474 (0.285) | 0.563 (0.283) |
| **Closeness** | −0.034 | 0.273 | < .025 | 0.408 (0.300) | 0.477 (0.272) |
| **Avg. Node Depth** | −0.259 | 0.020 | < .025 | 2.603 (1.854) | 2.239 (2.089) |
| | | | | **Zone 1** | **Zone 3** |
| **Parameter** | **Max. Neg** | **Max. Pos.** | **p-value** | **Mean (StDev)** | **Mean (StDev)** |
| **Outdegree** | −0.042 | 0.095 | > .100 | 0.474 (0.285) | 0.437 (0.261) |
| **Closeness** | −0.026 | 0.190 | > .100 | 0.408 (0.300) | 0.337 (0.259) |
| **Avg. Node Depth** | −0.165 | 0.029 | > .100 | 2.603 (1.854) | 3.292 (3.839) |
| | | | | **Zone 2** | **Zone 3** |
| **Parameter** | **Max. Neg** | **Max. Pos.** | **p-value** | **Mean (StDev)** | **Mean (StDev)** |
| **Outdegree** | −0.015 | 0.249 | < .025 | 0.563 (0.285) | 0.437 (0.261) |
| **Closeness** | −0.003 | 0.370 | < .001 | 0.477 (0.272) | 0.337 (0.259) |
| **Avg. Node Depth** | −0.261 | 0.003 | < .025 | 2.239 (2.089) | 3.292 (3.839) |

## 6  Conclusions

In this work the seismic clusters of South and South-eastern Spain are identified through the NN algorithm. Then, under the formalism of graph theory, the relations between the events of the clusters are used to both represent the cluster structures and compute the centrality measures, namely: the outdegree centralisation, the closeness centralisation, and the average leaf depth. These measures are useful towards the understanding of the tectonic behaviour and classification of the regions according to the complexity of the clustered seismicity (Peresan and Gentili, 2018, 2020; Talebi et al., 2024).

Three different models for the zonation have been proposed based on the Spanish ZESIS tectonic zonation units as building blocks. According to the Kolmogorov-Smirnov test, two out of the three proposed zonations (Zonation 1 and Zonation 2) reveal statistically significant differences between zones with respect to clustering characteristics. In other words, for each of the two zonations, the distributions of the considered scalar measures differ across zones. Given the results exposed in the previous section and the fact that Zonation 2 covers most of the investigated region and includes more events, and hence increases the statistics for clusters and background seismicity investigations, we consider it as the preferred option.

The eastern-most zone is characterised by prevailing burst-like structure for the clusters related to the External zone tectonic setting. The western-most zone, instead, exhibits a prevalent chain-like cluster structure (with relatively high complexity) that could be related to the Internal zone tectonic setting, which is characterised by dense fault system and where swarms are prone to happen.

The study shows that the clustering properties could help redefine the seismic zonation, particularly if a declustered catalogue is to be used in the seismic hazard assessment, as the regions that share the clustering properties might have a common background seismic activity rate due to their underlying tectonic setting.

Although the focus of this work is set on the study of the cluster structure properties and the classification of regions using the aforementioned complexity measures, it is worth noting that several studies have related the regional heat flow and fluid balance to swarm behaviour in seismicity and the occurrence of high magnitude earthquakes. For instance, Martínez-Garzón et al. (2018) found a positive correlation between the increase of geothermal activity (fluid balance) with the number of aftershocks in seismic series. Other works such as the one from Papadakis et al. (2016) relate relatively high heat flow values with strong earthquakes with focal depths lower than 40 km in Greece. In the area of study, Luque-Espinar and Mateos (2023) found notable changes in the geochemistry of thermal waters during the 2020-2021 Granada swarm including temperature changes. They highlighted the role of the variations in the $SO_4$ concentration as precursor; this signal increased during the seismic sequence peaking before the highest magnitude earthquakes occurred.

A future work could explore the use of these zonations to assess the seismic hazard by computing the Peak Ground Acceleration (PGA) for the areas that share similar clustering properties. These results should then be compared with those existing in the literature to see if they better represent the seismicity in each region. In addition, the possible temporal changes in background seismicity rates within the identified zones could be explored, following (Benali et al., 2020), so as to develop time-dependent hazard maps.

*Code availability.*  The code used in this study is available upon reasonable request.

## Appendix A:  Homogenisation of the Spanish seismic catalogue

Regarding the homogenization process, which can also affect the results, there are more than 6 types of magnitudes that can be found when the catalogue is downloaded after issuing a query in the IGN database webpage: https://www.ign.es/web/ign/portal/sis-catalogo-terremotos (IGN, 2022). Each local magnitude scale (https://www.ign.es/web/resources/docs/IGNCnig/SIS-Tipo-Magnitud.pdf) has a linear function that relates it to the moment magnitude (Mw) scale and a range of application (minimum and maximum local magnitude scale value) as seen in the last update of the Spanish Seismic Hazard Map (IGN-UPM Working Group, 2013). Table A1 summarises the equations for the four intensity/magnitude scales that appear in the area of study:

**Table A1.** Homogenisation equation for magnitude types present in the area of study. Obtained from IGN-UPM Working Group (2013, pp. 28-29).

| $y = a + b \cdot x$ | Range | Used range | Type |
|---|---|---|---|
| $Mw = 1.656 + 0.545 \cdot I_{max}$ | 3.0 - 9.5 | 2.0 - 5.0 | 1 |
| $Mw = 0.290 + 0.973 \cdot mbLg$ | 3.1 - 7.3 | 0.6 - 5.2 | 2 |
| $Mw = -1.528 + 1.213 \cdot mb$ | 3.7 - 6.3 | 3.0 - 5.5 | 3 |
| $Mw = 0.676 + 0.836 \cdot mbLg$ | 3.0 - 5.1 | 0.1 - 4.6 | 4 |

For this work, the above equations were used to convert the local magnitude and intensity measures to moment magnitude. As for the magnitude range of the conversions we used the one in the corresponding column given that the ranges are narrow for some of the conversion functions and following them would have resulted in a bias of the completeness magnitudes for the most recent periods.

IGN-UPM Working Group (2013, pp. 28-29) supplies the histograms with the difference between the computed magnitudes and the supplied local magnitudes, and it can be seen that for types 2 and 4, these differences are centred around 0. This is not the case in the type 3, which also shows a higher uncertainty. Plotting the conversion function (Figure A1) allows to see how type 2 and 4 magnitudes do not deviate greatly from the Mw. This is not the case for type 3, as the slope is higher for this conversion function.

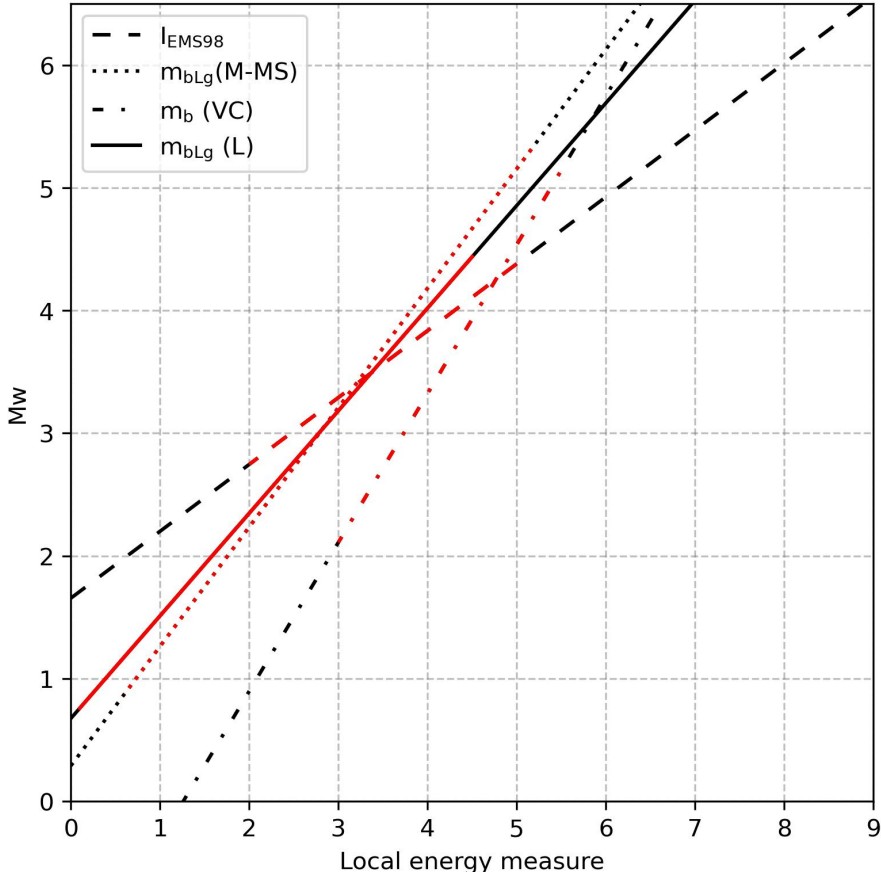

**Figure A1.** Example of application of the type 1, 2, 3 and 4 conversion to Mw from local energy measure. It can be seen that Type 3 slope is higher, so extrapolating this function to a wider range would result in high uncertainty. The red coloured section indicate the measure range as indicated in Table A1. The uncertainty in the conversion is shown as a coloured fill around the lines.

Figure A2 shows the event distribution for the whole catalogue depending on their magnitude type before conversion. Following the referee's question we have decided to add a new section in appendix, including this figure, so as to highlight the importance of the homogenisation step in the catalogue preparation. Type 5 comprises those events already in Mw in the catalogue.

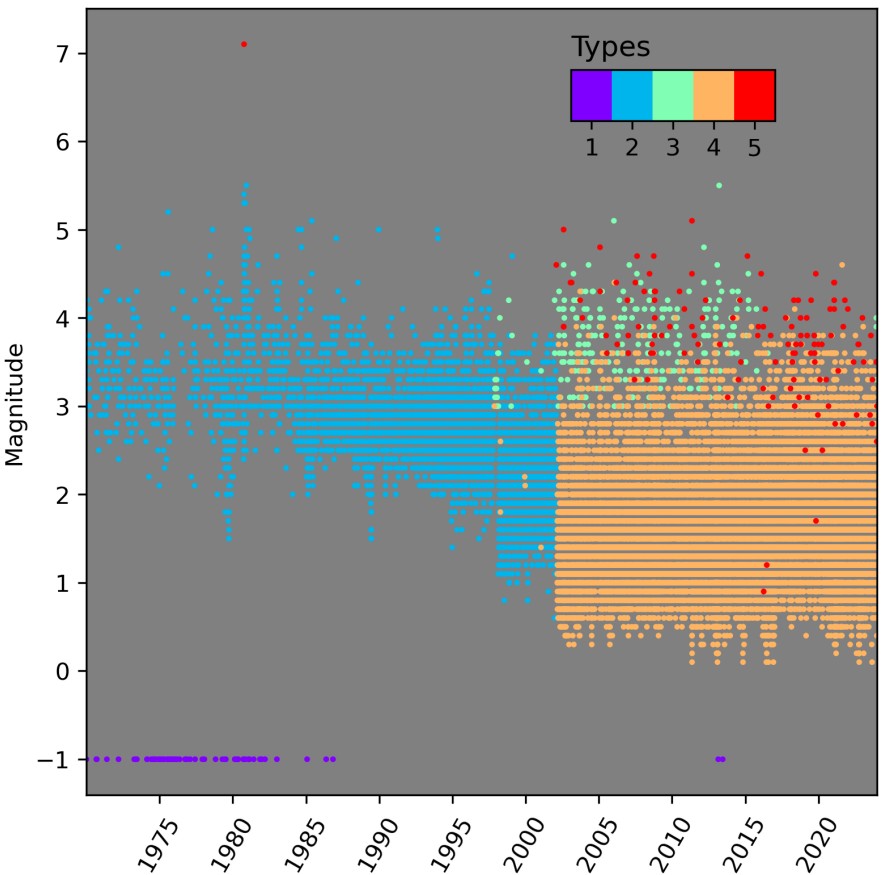

**Figure A2.** Magnitude type distribution for the catalogue before conversion to Mw. Magnitude equal to -1 has been assigned to those events with only intensity values in order to differentiate them.

## Appendix B: Scale parameters optimisation

In order to check the influence of these parameters in the cluster analysis, a variation in these has been used as input in the NN method. Table B1 compiles the set of parameters used in this work:

**Table B1.** Optimal sets of parameters to be used in further cluster structure analysis.

| Parameter | Minimum value | Computed value | Maximum value |
|:---:|:---:|:---:|:---:|
| **b-value** | 0.9 | $1.12 \pm 0.01$ | 1.3 |
| **Completeness magnitude, $M_c$ [Mw]** | 2.9 | $3.0 \pm 0.1$ | 3.2 |
| **Fractal dimension, d** | 1.4 | $1.5 \pm 0.1$ | 1.6 |

We use different statistics to evaluate the influence of these parameters on the results. For instance, regarding the spatial distribution of the events identified as a member of a cluster, the z-score for the parametric distribution of the distances from the rest of the events of the cluster to the mainshock has been computed:

$$z = \frac{d_{ik} - \tilde{d}_{ik}}{IQR} \tag{B1}$$

where $d_{ik}$ is the distance from the event $i$ to the mainshock of the cluster $k$, $\tilde{d}_{ik}$ is the median of these distances and $IQR$ is the interquartile range.

When a z-score is higher or equal than 2 then that event is regarded as a spatial anomaly.

In the case of the time distribution of the events, we have considered that when 6 months or more have passed from one event to the next one in the cluster, then the latter event and those after it are time anomalies. This approach is taken for those events with magnitudes lower than Mw4.0.

With these definitions the parameters' influence on the results will be evaluated in a preliminary analysis by computing the number of spatial and time anomalies in each cluster. The initial parameter configuration will correspond with the "Computed value" column in Table B1.

## B1 Influence of the b-value

Figure B1a shows that the influence of the b-value in the total number of both time and spatial anomalies is not clear. However, in Figure B1b it can be seen that the higher the b-value the higher the maximum number of anomalies in a cluster.

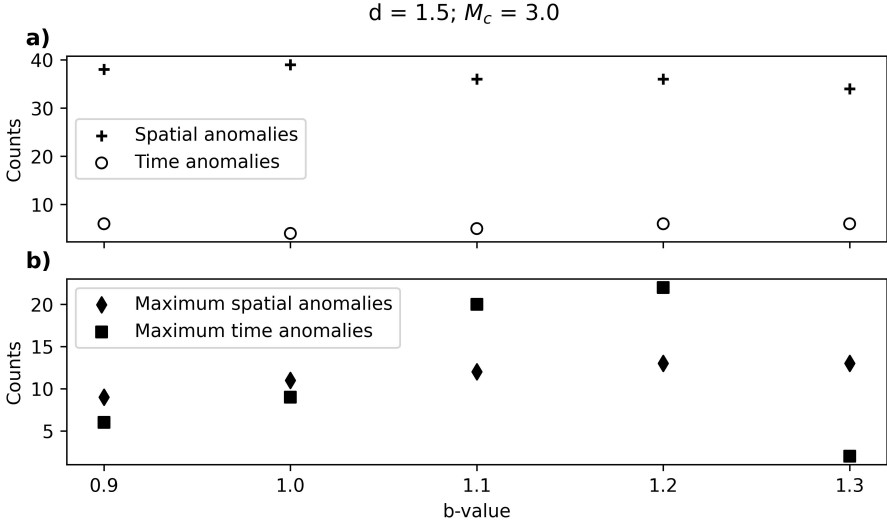

**Figure B1.** a) Clusters with spatial and time anomalies for different b-values and b) maximum spatial and time anomalies per cluster for different b-values.

## B2   Influence of the completeness magnitude

Figure B2a and B2b show that, in general, both the anomalies and maximum number of anomalies per cluster decrease with increasing completeness magnitude. This could be related with the fact that an increasing completeness magnitude effectively means less events in the catalogue.

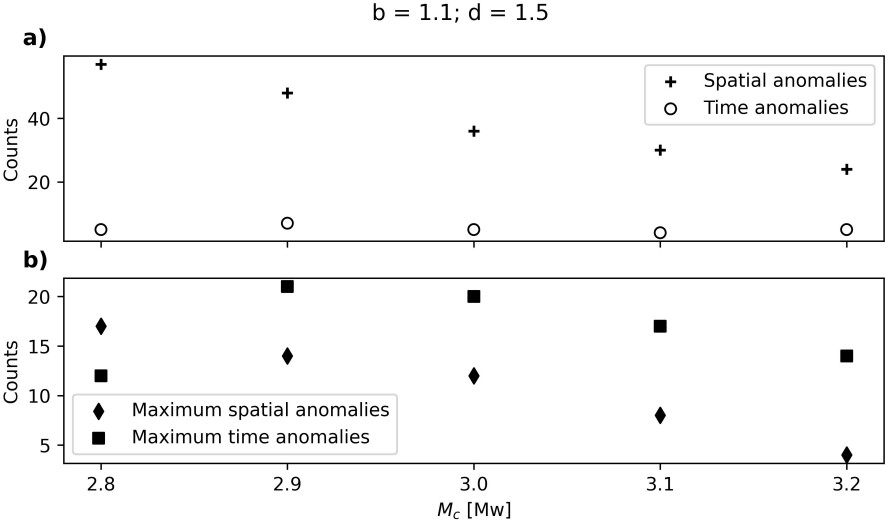

**Figure B2.** a) Clusters with spatial and time anomalies for different completeness magnitudes and b) maximum spatial and time anomalies per cluster for different completeness magnitudes.

## B3   Influence of the fractal dimension

As in the case of the b-value it can be seen in Figure B3a that the number of clusters with spatial and time anomalies does
not change significantly with the fractal dimension for the considered range. Nevertheless, the maximum number of spatial anomalies (Figure B3b) rises with the fractal dimension. This is directly related with (1) and the physical meaning of this parameter. A fractal dimension closer to 2 would mean the structure approaches to covering the whole 2D surface, which in turn involves greater spatial dispersion around the mainshocks.

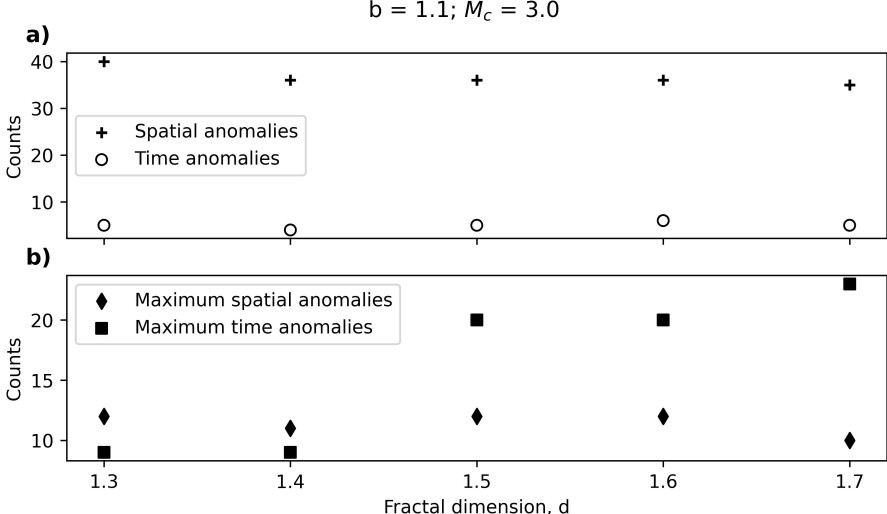

**Figure B3.** a) Clusters with spatial and time anomalies for different completeness magnitudes and b) maximum spatial and time anomalies per cluster for different completeness magnitudes.

Finally, the minimum distance to be considered in the NN distance algorithm will be analysed. In principle, the only con-
straint to this parameter is the epicentral uncertainty in the catalogue. This uncertainty has been already studied for some of
the periods the catalogue covers (González, Á., 2017) so it can be bounded by 10 km as the highest value and 2 km as lowest
(computed as the mean epicentral error of the catalogue from 2000 on). The tendency is not clear for the time anomalies as
seen in Figure B4a and B4b, but it seems to be optimal for 7.5 km in the case of the maximum spatial anomalies in Figure B4b.

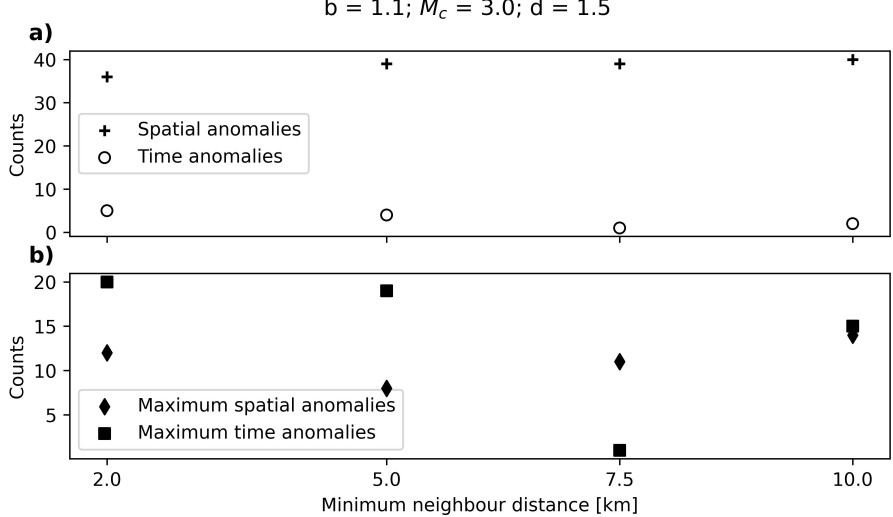

**Figure B4.** a) Clusters with spatial and time anomalies for different minimum neighbour distances and b) maximum spatial and time anomalies per cluster for different minimum neighbour distances.

## Appendix C: Anomaly study for the main set of parameters

Figure C1 and Figure C2 summarise the main anomaly analysis for these sets of parameters. Both sets of parameters minimise the number of clusters with time anomalies, although there is still one cluster with 10 or more anomalous events. As for the spatial anomalies, most of the clusters with spatial anomalies have only one anomaly.

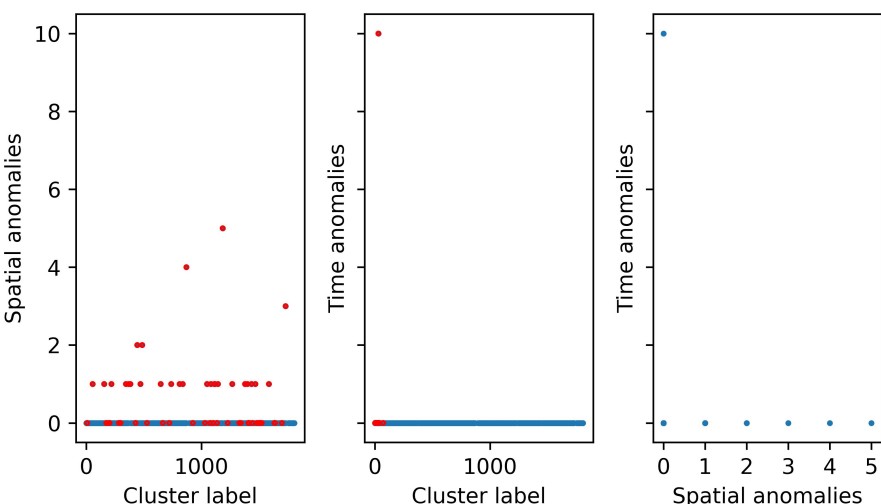

**Figure C1.** Anomaly analysis for the clusters identified with NN algorithm using the following set of parameters: b-value, 1.0; Completeness magnitude, Mw3.2; Fractal dimension, 1.5 and Minimum neighbour distance, 7.5 km. The red dots represent the clusters with 4 or more events.

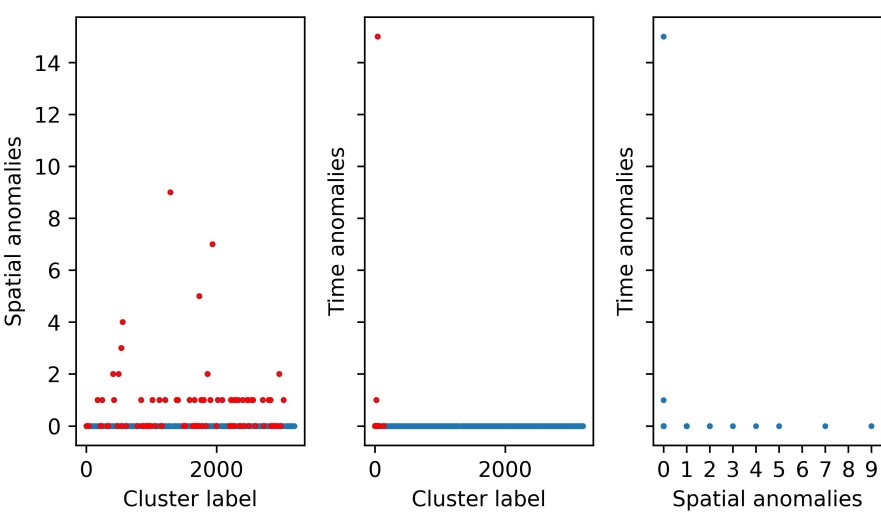

**Figure C2.** Anomaly analysis for the clusters identified with NN algorithm using the following set of parameters: b-value, 1.0; Completeness magnitude, Mw3.0; Fractal dimension, 1.5 and Minimum neighbour distance, 7.5 km. The red dots represent the clusters with 4 or more events.

## Appendix D: Anomaly study for the critical threshold, $\eta_0$

Figure D1 and Figure D2 compare the spatial and time anomalies in the clusters when using the fixed $\eta_0$ value of -4.5 for the two main set of parameters: 1) b-value, 1.0; Completeness magnitude, Mw3.2; Fractal dimension, 1.5 and Minimum neighbour distance, 7.5 km; and 2) b-value, 1.0; Completeness magnitude, Mw3.0; Fractal dimension, 1.5 and Minimum neighbour distance, 7.5 km. These results should be compared with those in the section Appendix C as they were obtained by using a free $\eta_0$ value ($\eta_0 = -3.4$ for the first set of parameters and $\eta_0 = -3.5$ for the second one).

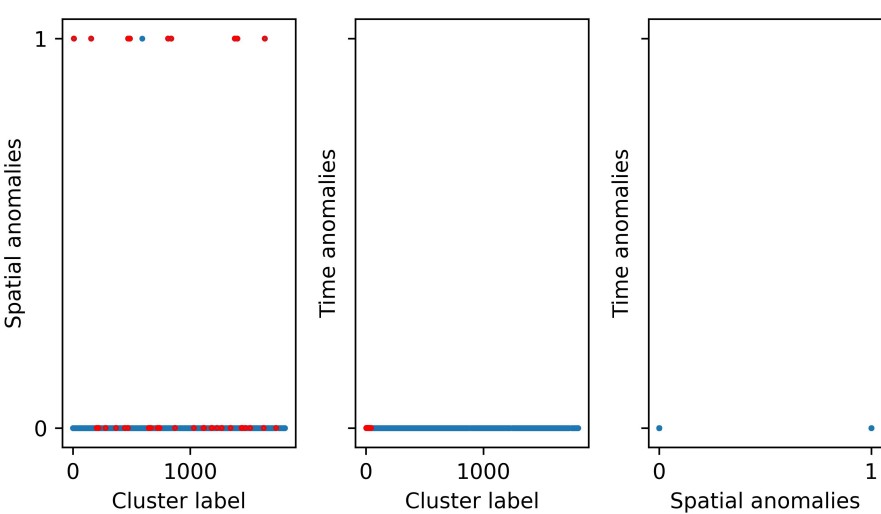

**Figure D1.** Anomaly analysis for the clusters identified with NN algorithm using the following set of parameters: b-value, 1.0; Completeness magnitude, Mw3.2; Fractal dimension, 1.5; Minimum neighbour distance, 7.5 km and critical threshold, $\eta_0$, -4.5. The red dots represent the clusters with 4 or more events.

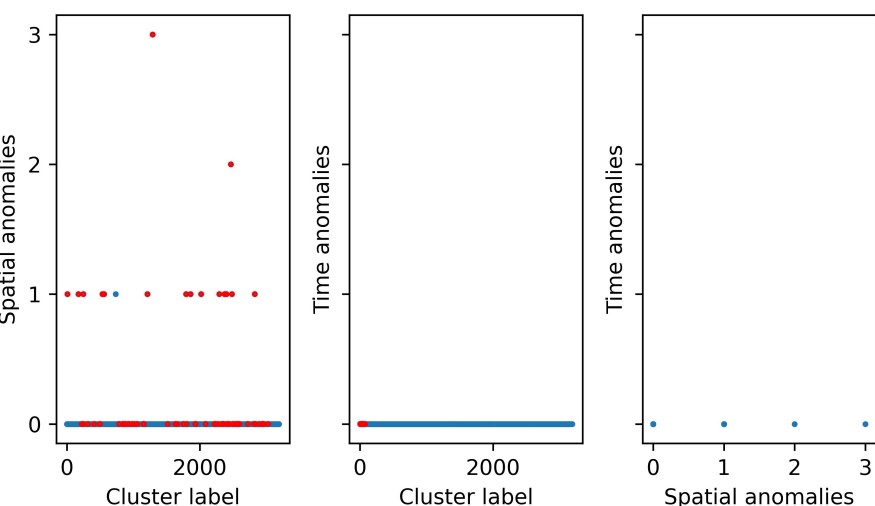

**Figure D2.** Anomaly analysis for the clusters identified with NN algorithm using the following set of parameters: b-value, 1.0; Completeness magnitude, Mw3.0; Fractal dimension, 1.5; Minimum neighbour distance, 7.5 km and critical threshold, $\eta_0$, -4.5. The red dots represent the clusters with 4 or more events.

*Author contributions.* Conceptualisation and original idea: AP and DML; methodology: DML, AP, EV and SM; EV wrote the code regarding the clustering properties, AP wrote the code for the statistical testing and DML wrote the code for the fractal dimension computation (box counting), spatial and temporal anomalies checking and catalogue analysis and plotting; DML, AP and SM performed the data curation; writing the original draft: DML, SM, AP and EV; writing review and editing: DML, SM, AP and EV. All authors have read and agreed to the published version of the manuscript.

*Competing interests.* The authors declare that they have no conflict of interest.

*Acknowledgements.* The authors thank the anonymous referee and Patricia Martínez Garzón for their insight, as it helped to improve this manuscript. DML and SM were partially supported by the European Regional Development Fund (FEDER), the Government of Spain (Ministry of Science and Innovation) through the reference project PID2021-123135OB-C21, by the Regional Government of Valencia (Ministry of Education, Culture, Universities and Employment) through the reference project: CIAICO/2022/038, and by the Research Group VIGROB-116 (University of Alicante). AP benefited from financial support from the RETURN Extended Partnership (Next-Generation EU).

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
