# Peer review of "Insights into tectonic zonation models from the clustering analysis of seismicity in South and South-eastern Spain"

_EGUsphere, 2025_

## Author Comment (AC1)

**FIRST REFEREE**

First, we thank the Anonymous referee for their insightful comments and questions as they were helpful towards the clarification and improvement of the manuscript. In the following document their questions are addressed in the same order as they were presented in the referee's report.

1. **The point "few earthquakes are associated with specific fault segments and occurrence time periods (when indicated) are affected by high uncertainties " in the abstract is not quite clear explained in introduction. It is suggested to supplement relevant content.**

   This sentence is a slightly modified quote to Gaspar-Escribano et al. (2015) "... data quality is very heterogeneous; there is only a few earthquakes associate to specific fault segments; and recurrence periods (when indicated) contain strong uncertainties." in the context of using the QAFI database to define fault seismic sources. In order to avoid any confusion, the page of the paper where the citation can be found has been added to the citation: (Gaspar-Escribano et al. 2015, p. 67)

   This consideration, nevertheless, can be extrapolated to certain recurrence times for historical earthquakes, as their magnitude is computed by homogenization of the intensity value, which as a subjective scale yields great uncertainties, as stated by the previous authors in section 2 first paragraph, page 63, in the context of catalogue uncertainties.

   This is what motivated the authors to base the seismic hazard analysis in polygonal sources rather than fault sources.

   The following sentence has been added in the four-to-last paragraph of section 1 to clarify this point:

   "*It is important to state that the high uncertainties in the QAFI database (Garcia-Mayordomo et al., 2012) and lack of earthquakes related to certain fault segments as pointed out by Gaspar-Escribano et al. (2015, p. 67) rules out using a fault based seismic source model.*"

2. **In section 3.1, Mc and b value is presented using Table 1 and Figure 5. These two contain a lot of duplicate information, it is recommended to merge. And the graphs and tables in the article are many, it is suggested to adjust appropriately and retain the more critical ones.**

   We agree with the referee in this regard. Since we wanted for the reader to be able to check the values, we presented the table as well, but in order to avoid duplicities we have modified figure 5 so it also shows the values

next to each marker in the graph. This way these values can be easily compared with other works.

New version of Figure 5 to be used so Table 1 is redundant:

[Figure]

*Figure 5. Changes in the b-value and completeness magnitude over time along with the evolution in the number of seismic stations near the area of study for the period 1970-2014. The markers represent the values of the b-value (blue) and completeness magnitude (red) from the year in which they are plotted to the next marker's year location. For example, the first marker would be the b-value (completeness magnitude) from 1970 to 1984. The dashed lines indicate the values of the parameters for the whole catalogue (1970-2023).*

3. **From Table 2, it is observed that the two datasets exhibit a difference of 0.2 in the Mc value; however, there is a significant disparity in the number of recorded earthquakes. Are there any additional differences between the two earthquake catalogs that could account for this variation?**

In both cases the original catalogue is the same, the difference between the two datasets appears when the filtering is applied for completeness magnitude Mw3.0 and Mw3.2.

In the following table we present the size of the catalogue depending on the completeness magnitude for the area of study.

| Completeness magnitude | Total number of events in the catalogue |
|:---:|:---:|
| 3.0 | 3191 |
| 3.2 | 1806 |

This analysis can be extrapolated to lower magnitudes:

[Figure]

*Figure.  Cumulative number of earthquakes per year, for different moment magnitude thresholds. The black vertical lines indicate the times of completeness magnitude changes.*

This behaviour is not exclusive of the Mw3.0-Mw3.2 bin, it can be seen in other bins as well. This tendency could be related to the seismicity in the area (periods of more seismic activity in the regions, for example during 1990). Another important factor could be the inequal development of the seismic network, which causes heterogeneity in the completeness magnitude (meaning there is an excess in a magnitude bin in the energy-frequency histogram due to detection limits). This can be also seen in Figure 5 where the number of stations almost doubles from 1990s to 2000s.

1. **In Section 4.1, the metrics of outdegree centralization, closeness centralization, and average leaf depth are utilized**

**to assess the characteristics of clusters. Additionally, various types of seismicity are mentioned, including burst-like, swarm-like, chain-like, and umbrella-like phenomena. It is recommended to unify these terms, as some possess similar meanings, to enhance clarity and coherence. Furthermore, a more detailed explanation of the specific characteristics of each cluster type would provide a deeper understanding of the underlying seismic activities.**

We agree with the referee that adding more context to the explanation could benefit the manuscript. The terms umbrella-like/burst-like indeed refer to the same cluster structure, the same way chain-like/swarm-like do. From sections 4.2 on we prefer using the terms burst-like and swarm-like as they can be more easily related to the kind of cluster to be expected. Nevertheless, in the section 4.1 we present both naming options in order to explain the graph theory concepts as they can appear in several other papers.

In order to clarify what these terms mean we provided an additional figure and the interpretation of Figures 8 and 10 (for Adra Sequence and Granada swarm) using this classification.

"Figure 12 shows an example of such cluster structure types according to their tree graphs. In reality, more complex cluster structures are expected due to combination of these typologies. For instance, Adra's sequence (Figure 8) could be classified as a double-umbrella-like cluster whereas Granada's swarm (Figure 10) has a more pronounced chain-like component."

[Figure]

*Figure 12. Cluster types according to their tree-graph structures. The events are represented with sizes and grey-shade colours according to their magnitude.*

4. **Figure 3 presents the magnitude-depth distribution of seismic events. It would be interesting to further investigate whether the differences in seismic activity characteristics between tectonic regions can be better understood through a combined analysis of focal depths and focal mechanisms. This could be a direction for future research.**

We agree that the analysis of focal depth and mechanism can benefit both the understanding of the seismic activity behaviour in the different tectonic zones and help identifying and characterizing the seismic sources. This would require a re-evaluation of the focal depths in all the Spanish catalogue, as the uncertainty in this parameter has changed over time (as well as in the case of the epicentral location) and should be accounted for. For this task, a better knowledge on the fault planes is needed, as in such cases in which no clear solution for the focal depth is obtained, one could use the parameters in the fault plane and, through methods such as Monte-Carlo, simulate the most likely depth for the hypocentre.

With the combined information of both focal depths, focal mechanisms and cluster structures (and fault trace location and plane geometry), one possible direction of research could focus on analysing the evolution of the seismic series considering induced and auto-induced seismicity in complex faulting scenarios such as in the case of Granada.

---

## Author Comment (AC2)

**SECOND REFEREE**

We are grateful to Dr. Patricia Martínez Garzón for her valuable comments, suggestions and questions as they were helpful and contributed to the improvement of the manuscript. The questions are answered below in the original order.

**1# On the *declustering* of the seismicity catalog: The authors discuss in the introduction the importance of declustering the catalog to study seismic hazard analysis. However, to my understanding, in the following, the paper mostly focuses on the analysis of the seismicity clusters, rather than on the background (*declustered*) catalog. Thus, it is not clear what is the main purpose of the manuscript (e.g. the declustered seismicity, or the analysis of clusters). Depending on which one, the nearest-neighbor strategy to be applied is different (i.e. b=0, see Zaliapin and Ben-Zion, 2021 for this topic). See for example the name of the section 3.2 "Role of Nu in the declustering". I believe "Role of Nu in the identification of clusters" would better represent the goals of the paper.**

We agree with the referee. This manuscript aims to study the clustering properties of seismicity in this region and to evaluate the possibility of using cluster complexity measures to differentiate tectonic zones (redefine or group). These redefined zones can then be used to perform the seismic hazard analysis. In this sense, although it is not the main objective, the catalogue declustering is a "by-product" much needed for the seismic hazard analysis of the background seismicity. The 3.2 section title has been changed as we think the provided suggestion fits the focus of the manuscript. We thank the referee for this valuable comment.

**2# On the *homogeneity of magnitude types*: Can the authors confirm that all employed magnitudes correspond to Moment Magnitudes $M_w$? Typically, regional catalogs employ rather local magnitudes, and scaling conversions need to be applied. This could affect these results and would be important to confirm it.**

We agree that the homogenization of the catalogue is an important step that should be followed when studying the seismicity of an area. The use of $M_w$ ensures that the catalogue can be shared and used for wide variety of purposes, as it can be easily related with the released energy without having to deal with site effects/equipment considerations.

Regarding the homogenization process, which can also affect the results, there are more than 6 types of magnitudes that can be found when the catalogue is downloaded after issuing a query in the IGN database webpage (https://www.ign.es/web/ign/portal/sis-catalogo-terremotos). Each local magnitude scale ( https://www.ign.es/web/resources/docs/IGNCnig/SIS-Tipo-Magnitud.pdf ) has a linear function that relates it to the moment magnitude

($M_w$) scale and a range of application (minimum and maximum local magnitude scale value) as seen in the last update of the Spanish Seismic Hazard Map (IGN-UPM Working Group, 2013).

| y = a + bx | Range | Used range | Type |
|---|---|---|---|
| Mw = 1.656 + 0.545·Imax | 3.0 - 9.5 | 2.0 – 5.0 | 1 |
| Mw = 0.290 + 0.973·mbLg | 3.1 - 7.3 | 0.6 – 5.2 | 2 |
| Mw = -1.528 + 1.213·mb | 3.7 - 6.3 | 3.0 - 5.5 | 3 |
| Mw = 0.676 + 0.836·mbLg | 3.0 - 5.1 | 0.1 – 4.6 | 4 |

For this work, we used these equations to convert the local magnitude and intensity measures to moment magnitude. As for the magnitude range of the conversions we used the one in the corresponding column given that the ranges are narrow for some of the conversion functions and following them would have resulted in a bias of the completeness magnitudes for the most recent periods.

[Figure]

Figure 1. Example of application of the type 2, 3 and 4 conversion to Mw from(?) a local magnitude ranging between 0 and 7. It can be seen that Type 3 conversion has a steepest slope. Therefore, extrapolating this function beyond the given range would introduce significant uncertainty. The red sections of each line correspond to the used ranged for the conversion in this work.

IGN-UPM Working Group (2013, pp. 28-29) supplies the histograms with the difference between the computed magnitudes and the supplied local magnitudes, and it can be seen that for types 2 and 4, these differences are centred around 0. This is not the case in the type 3, which also shows a higher uncertainty.

The next figure shows the event distribution for the whole catalogue depending on their magnitude type before conversion. Following the referee's question we have decided to add a new section in appendix, including this figure, so as to highlight the importance of the homogenisation step in the catalogue preparation. Type comprises those events already in Mw in the catalogue.

[Figure]

*Figure 2. Magnitude type distribution for the catalogue before conversion to Mw. Magnitude equal to -1 has been assigned to those records with only intensity values.*

**3# *On the two datasets from Table 2*: I did not understand what is the purpose of this separation of datasets, I believe that there is some mistake in the table, as the number of events included is dramatically different, but the parameters specified in Table are almost the same. Even if included in the Supp. Materials, a minimum of 1-2 lines are needed to justify the employment of these datasets and what they represent.**

In this section the purpose is to analyse the cluster structure by first identifying the members of each cluster. For this reason, we selected the overall completeness magnitude (for the whole catalogue in this area and from 1970 on) at around Mw3.0 and the maximum completeness magnitude Mw3.2 for the period from 1970 to 1984 (the most restrictive condition). We agree that this matter should be explained in the text where the table is presented. As pointed out, the difference between the number of events in each dataset is notable despite the magnitude difference being Mw0.2. This is not exclusive of the 3.0-3.2 Mw bin, it can be observed for other bins such as Mw2.6-2.8, Mw2.8-3.0.

[Figure]

*Figure. Cumulative number of earthquakes per year, for different moment magnitude thresholds. The black vertical lines indicate the times of completeness magnitude changes as computed in this work.*

This behaviour may be explained by observing the number of stations per year, shown in Figure 5 in the manuscript (now modified):

[Figure]

*Figure 5. Changes in the b-value and completeness magnitude over time along with the evolution in the number of seismic stations near the area of study for the period 1970-2014. The markers represent the values of the b-value (blue) and completeness magnitude (red) from the year in which they are plotted to the next marker's year location. For example, the first marker would be the b-value (completeness magnitude) from 1970 to 1984. The dashed lines indicate the values of the parameters for the whole catalogue (1970-2023).*

It can be seen that from the late 1980s to 2000s the number of seismic stations in the network doubles, which could explain the steep increase in the number of detected events. Which could also explain the great decrease in the completeness magnitude within the period from 1985-1998 to 1999-2013.

**4# On the analysis of clusters: It is not specified whether for estimating the rescaled distances, authors are using hypocentral or epicentral distance. This is important to understand which fractal dimension should be used.**

Thank you for bringing this up as it should be specified, being one of the parameters used in the computation of the rescaled distance. As can be seen in Figure 3 in the manuscript (attached below), most of the seismicity is concentrated in the first 10 km of depth in the crust. For this reason, as well as due to the large uncertainty affecting depth estimates, we decided using the epicentral distance rather than the hypocentral distance. It should also be noted

that the depth in the oldest part of the catalogue is more affected by uncertainty, making the use of the depth in the hypocentral distance computation unreliable.

Due to the large uncertainties in hypocentral depth determination, following Zaliapin and Ben Zion (2020) the analysis is performed considering epicentral coordinates, and thus is not affected by the depth uncertainties. When dealing with global scale analysis, where very deep earthquakes are reported, events are eventually selected within specified depth ranges (e.g. Zaliapin and BenZion, 2016), while for narrow scale studies, where reliable depth information is available, it can be taken into account (e.g. Martinez-Garzon et al., 2018).

This explanation has been added to the corresponding section of the manuscript.

[Figure]

*Figure 3. Magnitude-Depth distribution for the chosen catalogue. The histogram in the inset shows the depth distribution.*

**5# Metrics used: Returning to the issue of analysis of clusters vs declustering: the metrics defined in section 4.1 solely analyze the clustered part of the seismicity catalog. I strongly suggest to include some other metrics that also analyze the background seismicity of the region. Some suggestions include the ratio of clustered to background seismicity, or the proportion of single seismicity on these regions (see e.g. Martínez-Garzón**

**et al., 2018; 2019 for some example of these metrics considering also the background seismicity).**

We are grateful to the referee for the suggestion; we have added a column to the former Table 3 (now 2) in section 4.3 with the clustered to background ratio. Also, a couple of lines have been added to the paragraph preceding the table:

"*It can be seen that the clustered to background seismicity ratio increases as the completeness magnitude decreases. This is to be expected, as lower magnitude earthquakes are more frequently associated with seismic sequences in the aftershock category. It is also in agreement with the increase in the ratio of aftershocks (17% for the Mw3.2 completeness magnitude vs 25% for the Mw2.1 completeness magnitude).*"

| | | | | Full catalogue | | |
|---|---|---|---|---|---|---|
| | | | | | Clusters | |
| Mc | n events | CtoB ratio | Singles | Foreshocks | Mainshocks | Aftershocks |
| 3.2 | 1806 | 0.43 | 1,262 (70%) | 92 (5%) | 149 (8%) | 303 (17%) |
| 2.1 | 20,057 | 0.64 | 12,210 (61%) | 1,244 (6%) | 1,643 (8%) | 4,960 (25%) |

**CtoB**: Clustered seismicity to Background seismicity ratio.

*Table 2. Summary of the clustering statistics for the two different completeness magnitudes.*

**6# On the distinction between burst and swarms: In fact, I don't see this distinction very clear, other than in the Adra vs Granada sequences. To check the potential bimodality of the clusters, I recommend to plot average leaf depth as a function of cluster size, similar to what done in Figure 2d from Martínez-Garzón et al., 2019. This should give us a better perspective if indeed it is related to the larger magnitude size, or if there might be physical reasons promoting bursts (here called umbrellas) or swarms.**

We agree with the reviewer regarding the need to clearly and univocally define the cluster types and related names; the scheme below has been added in order to clarify the general features of cluster types in the context of graph theory.

[Figure]

*Figure 12. Cluster types according to their tree graph structures. The events are represented with sizes and grey-shade colours according to their magnitude.*

We also agree that by plotting average leaf depth as function of cluster size could be useful towards a better understanding of the clustering properties, more specifically in the cluster type (according to the tree graph representation and the related measures of centrality).

As for the results in this manuscript, an Average leaf depth vs cluster size plot (for clusters with size 5 or more) has been added to section 5 along with the following paragraph:

"*It is interesting to compare then the results for both completeness magnitude thresholds in terms of average node depth and cluster size (Figure 15), as this representation can provide some insights on the possible relation between the cluster size and their different types. It can be observed that, especially for Mc=2.1, the average node depth <d> naturally increases with the cluster's size for swarm-like sequences, while for burst-like sequences <d> remains quite low even for large clusters, composed by more than a hundred events. Moreover, considering the geographical information (longitude values) it can be seen that in the west zone (longitude < -3º) complex clusters (higher average node depth) are more common than in the east zone, even for similar cluster sizes.*"

[Figure]

*Figure 15. Average leaf depth vs cluster size for completeness magnitude 3.2 (left) and completeness magnitude 2.1 (right). The red markers are events at the west side of the area of study (longitude < -3º) and those blue-coloured belong to the east side (longitude > -3º).*

**7# Finally, although this paper does not focus on understanding the physical processes, it would be good to note somewhere in the paper than the behaviour of bursts vs swarms has been mainly related to the heat flow as well as the content of fluids in the crust. I think it may be worth to connect and refer to papers characterizing these processes in the here analyzed region.**

The aim of this study is to characterize the clustering properties to highlight the importance of such studies in the context of a seismic hazard analysis: be it the background seismic hazard (for the background seismicity) or the seismic hazard related to an aftershock sequence (namely to better understand how the seismic hazard may evolve during a seismic sequence). Still, we agree with this suggestion, and we find it useful to include in the conclusions a paragraph explaining which may be the cause (or important factors) that can be related to these different cluster structures, as follows.

*"Although the focus of this work is set on the study of the cluster structure properties and the classification of regions using the aforementioned complexity measures, it is worth noting that several studies have related the regional heat flow and fluid balance to swarm behaviour in seismicity and the occurrence of high magnitude earthquakes. For instance, Martínez-Garzón et al. (2018) found a positive correlation between the increase of geothermal activity (fluid balance) with the number of aftershocks in seismic series. Other works such as the one from Papadakis et al. (2016) relate relatively high heat flow values with strong earthquakes with focal depths lower than 40 km in Greece. In the area of study, Luque-Espinar and Mateos (2023) found notable changes in the geochemistry of thermal waters during the 2020-2021 Granada swarm including temperature changes. They highlighted the role of the variations in the $SO_4$ concentration as precursor; this signal increased during the seismic sequence peaking before the highest magnitude earthquakes occurred".*

I hope the authors find these comments useful.

We are grateful to the reviewer for the valuable comments, which allowed us to clarify some important elements of our analysis.

Regards,

Patricia Martínez-Garzón

Other comments:

**Lines 35-and after: Please indicate these notable earthquakes in Fig. 1 or in Fig. 2.**

Figures 1 and 2 now also show the indicated notable earthquakes. The caption on the figures also references this fact. The figures have been added in the following pages of the response (as some other modifications have been applied).

**Line 83-85: We need to know a bit more details on this work about the splitting of the catalog on these four periods, without having to consult Gonzalez (2017) for a minimum info. "It can be seen clearly from Figure 4", in fact I cannot see this so clearly.**

The paragraph preceding Figure 4 has been modified in order to be clear and also relating this subdivision of the catalogue with Figure 5 as well.

*"The detection sensitivity and, therefore, the completeness magnitude ($M_c$) of the catalogue have changed over time due to upgrades in the seismic network. This fact is thoroughly discussed in the work of González, A. (2017), the data from which has made possible identifying four periods with distinct seismic network sensitivity: 1970-*

*1984, 1985-1998, 1999-2013, and 2014-2023. These periods are also evident in Figure 5 by analysing the slope changes in the cumulative number of stations per year. Figure 4 shows the events of the Spanish catalogue for the area of study. The number of events with magnitudes lower than 3.0 increased from 1970-1984 to 1985 on, and then the number of events with magnitudes lower than 2.0 spiked from 1999 on, reflecting an improvement in the sensitivity of the Spanish seismic network."*

**Line 102: I think the correct references should be Zaliapin and Ben-Zion (2013a; 2013b), where the bimodality of the distributions is presented for first time.**

We thank the referee as Zaliapin and Ben-Zion (2013a) is the first instance in which the bimodality is discussed for the rescaled spatio-temporal distance distribution in the context of the real seismicity (section 3.3), which is used to identify the background and clustered events. The line has been corrected with the correct reference.

**Line 107: Labels (-1,1 and 2) are not needed in the paper.**

Agreed, this notation is only relevant to explain the labels obtained after applying Nearest-Neighbor algorithm and even in this case it also depends on the code implementation so it should be avoided.

**Line 109: I am not 100% sure but I think the correct term for "founder" in the Z&BZ nomenclature is *parent*.**

The line has been changed to: "*parent".*

**Section 4.1: The term "umbrella-cluster" is repeatedly used throughout the text, but it is not a well defined concept. I suggest to remove it from the text, as I feel that it is the same than the burst-like cluster topology defined by Z&BZ (2013b).**

Yes, we used it in relation to the shape of the tree graph of the cluster. We agree that is not found in the bibliography, so it has been removed from the text (or changed into burst-like when applicable) so it is easier to relate with other relevant works. We only kept the first reference of the shape (but not as an alternative name, in order to help illustrate the shape).

"*referred as burst-like (with an umbrella-like shape) and swarm-like (with a chain-like tree graph shape) sequences*"

**Figure 1: please add if possible stations to this figure, color encoded with the time of start operation.**

Done, Figure 1 now shows the stations in the area of study colour coded according to the year of first pick. The data has been obtained from the work of González (2017).

[Figure]

Figure 1. Main geological domains of South and South-eastern Spain (adapted from Buforn et al. (1995)). The red-edged dotted-filled polygons identify the Internal zone whereas the blue-edged polygons with a strip pattern fill mark the location of the External zone. The coloured stars represent the most damaging earthquakes in the area since the pre-instrumental era of the catalogue. The triangles represent the seismic stations' location, obtained from González (2017), and colour coded by the first year of operation.

**Figure 2: As M < 2 is below Mc during the entire time period, I suggest to remove them to try to clean a bit the figure.**

Agreed, it should have been done (in sort of retroactively way, considering that this magnitude is not used in the posterior analysis). We thank the referee as it is indeed hindering the visualization of the plot.

[Figure]

*Figure 2. Catalogue of South and South-eastern Spain from 1970 to the end of 2023. It can be seen that faulting system determines the location of the epicentres for the most relevant earthquakes (Mw between 5.0 and 6.0 and marked as red stars) as most of them are located near these structures. The fault traces have been obtained from the QAFI database (García-Mayordomo et al., 2012; IGME, 2022) and the tectonic zonation polygons from the ZESIS database (IGME, 2015). The coloured stars represent the most damaging earthquakes in the area since the pre-instrumental era of the catalogue.*

**Please also include the plot of the distribution of rescaled times and distances, preferentially as part of Figure 7.**

Sure, they have been added as part of the subplot in Figure 7.

[Figure]

*Figure 7c. Joint 2D distribution for the Rescaled time and rescaled space for dataset 1.*

[Figure]

*Figure 7d. Joint 2D distribution for the Rescaled time and rescaled space for dataset 2.*

The resulting plot in the manuscript is the following:

[Figure]

(a) First dataset

(b) Second dataset

(c) First dataset

(d) Second dataset

Figure 7. Histogram of the NN distances for (a) the first dataset and (b) the second dataset. Mixture model of two Gaussian density functions, one for background seismicity (yellow line) and the other for clustered seismicity (orange line), fitted to the datasets. Then, (c) and (d) show the 2D joint distribution of rescaled time and space for both datasets. The threshold distance η0 given by the estimated values −3.4 in the panel (a) and (c) and −3.5 in panel (b) and (d) (solid vertical line) or given by the fixed value −4.5 (dashed vertical line in all the panels).

**Figure 9: Please add the name of main fault structures and cities to these maps.**

Done, the names of the faults have been reduced to acronyms when possible. The explanation of which can be found in the caption of the figures. Only the locations with population greater than 30.000 (which can be considered as cities) have been plotted.

■ A: Adra Fault

■ AF: Alfacar Fault

■ AJ: Alitaje Fault

- ALB: Albuñuelas Fault

- AT: Atarfe Fault

- B-A: Belicena-Alhendín Fault190

- BA: Balanegra Fault

- BZ: Baza Fault

- CA: Carboneras Fault

- CU: Cubillas Fault

- D: Dílar Fault

- E: Escóznar Fault

- EGRFS: Eastern Gador Range Fault System

- F-J: El Fargue-Jun Fault

- G-L FZ: Graena-Lugros Fault Zone

- GR: Granada Fault

- GU: East of Guadix Fault

- H: Huenes Fault

- LdV: Loma del Viento Fault

- O-PP: Obéilar - Pinos Puente Fault

- P-N: Padul-Nigüelas Fault

- PP: Pinos Puente Fault

- PR: Pedro Ruiz Fault

- SdZFZ: Solana de Zamborino Fault Zone

- SF: Santa Fe Fault

- T-O: Tocón-Obéilar Fault

- VdZ: Ventas de Zafarraya Fault

**Mw5.2 23-12-1993 14:22:35UTC**

[Figure]

Figure 9. Adra's sequence, whose mainshock occurred on December 23, 1993, Mw5.2. Spatial distributions of the earthquakes' epicentres in the clusters obtained from the different set of parameters and critical threshold values: Top left: first dataset, estimated $\eta_0 = -3.4$; Top right: first dataset, tuned $\eta_0 = -4.5$; Bottom left: second dataset, estimated $\eta_0 = -3.5$; Bottom right: second dataset, tuned $\eta_0 = -4.5$. The central inset shows the location of the cluster's mainshock in Spain and the purple lines in each of the subplots mark the position of the active faults from QAFI v4.0 (García-Mayordomo et al., 2012; IGME, 2022) in the area. The blue labels indicate the different confirmed fault names and the black labels with the inverted triangle mark the position of the main cities in the area (population greater than 30,000).

[Figure]

Figure 11. Granada's swarm, whose mainshock occurred in January 23, 2021, Mw4.4. Spatial distributions of the earthquakes' epicentres in the clusters obtained from the different set of parameters and critical threshold values: Top left: first dataset, estimated η0 = −3.4; Top right: first dataset, tuned η0 = −4.5; Bottom left: second dataset, estimated η0 = −3.5; Bottom right: second dataset, tuned η0 = −4.5. The central inset shows the location of the cluster's mainshock in Spain and the purple lines in each of the subplots mark the position of the active faults from QAFI v4.0 (García-Mayordomo et al., 2012; IGME, 2022) in the area. The blue labels indicate the different confirmed fault names and the black labels with the inverted triangle mark the position of the main cities in the area (population greater than 30,000).

**References**

Leptokaropoulos, K., Staszek, M., Lasocki, S., Martínez-Garzón, P., & Kwiatek, G. (2018). Evolution of seismicity in relation to fluid injection in the North-Western part of The Geysers geothermal field. *Geophysical Journal International*, *212*(2), 1157–1166. https://doi.org/10.1093/gji/ggx481

Martínez-Garzón, P., Ben-Zion, Y., Zaliapin, I., & Bohnhoff, M. (2019). Seismic clustering in the Sea of Marmara: Implications for monitoring earthquake processes. *Tectonophysics*, *768*, 228176. https://doi.org/10.1016/j.tecto.2019.228176

Zaliapin, I., & Ben-Zion, Y. (2013a). Earthquake clusters in southern California I: Identification and stability. *Journal of Geophysical Research: Solid Earth*, *118*(6), 2847–2864. https://doi.org/10.1002/jgrb.50179

Zaliapin, I., & Ben-Zion, Y. (2013b). Earthquake clusters in southern California II: Classification and relation to physical properties of the crust. *Journal of Geophysical Research: Solid Earth*, *118*(6), 2865–2877. https://doi.org/10.1002/jgrb.50178

Zaliapin, I., & Ben-Zion, Y. (2021). Perspectives on Clustering and Declustering of Earthquakes. *Seismological Research Letters*, *93*(1), 386–401. https://doi.org/10.1785/0220210127

Papadakis, G., Vallianatos, F., & Sammonds, P. (2016). Non-extensive statistical physics applied to heat flow and the earth-quake frequency–magnitude distribution in Greece. *Physica A: Statistical Mechanics and its Applications,* 456, 135–144. https://doi.org/10.1016/j.physa.2016.03.022

Luque-Espinar, J. A. & Mateos, R. M. (2023). Hydrochemical changes in thermal waters related to seismic activity: The case of the 2020–2021 seismic sequence in Granada (S Spain), *Journal of Hydrology*, 627, 130. https://doi.org/10.1016/j.jhydrol.2023.130390